# FABind: Fast and Accurate Protein-Ligand Binding

**Qizhi Pei**[1,5]*, **Kaiyuan Gao**[2]*, **Lijun Wu**[3]†, **Jinhua Zhu**[4], **Yingce Xia**[3],
**Shufang Xie**[1], **Tao Qin**[3], **Kun He**[2], **Tie-Yan Liu**[3], **Rui Yan**[1,6]†

[1] Gaoling School of Artificial Intelligence, Renmin University of China
[2] School of Computer Science and Technology, Huazhong University of Science and Technology
[3] Microsoft Research AI4Science
[4] School of Information Science and Technology, University of Science and Technology of China
[5] Engineering Research Center of Next-Generation Intelligent Search
and Recommendation, Ministry of Education
[6] Beijing Key Laboratory of Big Data Management and Analysis Methods
{qizhipei,shufangxie,ruiyan}@ruc.edu.cn, {im_kai,brooklet60}@hust.edu.cn
{lijuwu,yinxia,taoqin,tyliu}@microsoft.com, teslazhu@mail.ustc.edu.cn

## Abstract

Modeling the interaction between proteins and ligands and accurately predicting their binding structures is a critical yet challenging task in drug discovery. Recent advancements in deep learning have shown promise in addressing this challenge, with sampling-based and regression-based methods emerging as two prominent approaches. However, these methods have notable limitations. Sampling-based methods often suffer from low efficiency due to the need for generating multiple candidate structures for selection. On the other hand, regression-based methods offer fast predictions but may experience decreased accuracy. Additionally, the variation in protein sizes often requires external modules for selecting suitable binding pockets, further impacting efficiency. In this work, we propose FABind, an end-to-end model that combines pocket prediction and docking to achieve accurate and fast protein-ligand binding. FABind incorporates a unique ligand-informed pocket prediction module, which is also leveraged for docking pose estimation. The model further enhances the docking process by incrementally integrating the predicted pocket to optimize protein-ligand binding, reducing discrepancies between training and inference. Through extensive experiments on benchmark datasets, our proposed FABind demonstrates strong advantages in terms of effectiveness and efficiency compared to existing methods. Our code is available at Github[3].

## 1 Introduction

Biomolecular interactions are vital in the human body as they perform various functions within organisms [44, 2]. For example, protein-ligand binding [35], protein-protein interaction [17], protein-DNA interaction [18], and so on. Among these, drug-like small molecules (ligands) binding to protein is important and widely studied in interactions as they can facilitate drug discovery. Therefore, molecular docking, which involves predicting the conformation of a ligand when it binds to a target protein, serves as a crucial task, the resulting docked protein-ligand complex can provide valuable insights for drug development.

---

*Equal contribution. This work was done during their internship at Microsoft Research AI4Science.
†Corresponding authors: Rui Yan (ruiyan@ruc.edu.cn) and Lijun Wu (lijuwu@microsoft.com).
[3]https://github.com/QizhiPei/FABind

37th Conference on Neural Information Processing Systems (NeurIPS 2023).

Though important, fast and accurately predicting the docked ligand pose is super challenging. Two families of methods are commonly used for docking: sampling-based and regression-based prediction. Most of the traditional methods lie in the sampling-based approaches as they rely on physics-informed empirical energy functions to score and rank the enormous sampled conformations [8, 35, 43], even with the use of deep learning-based scoring functions for conformation evaluation [34, 49], these methods still need a large number of potential ligand poses for selection and optimization. DiffDock [7] utilizes a deep diffusion model that significantly improves accuracy. However, it still requires a large number of sampled/generated ligand poses for selection, resulting in high computational costs and slow docking speeds. The regression-based methods [9, 32, 11] that use deep learning models to predict the docked ligand pose bypass the dependency on the sampling process. For instance, TankBind [31] proposes a two-stage framework that simulates the docking process by predicting the protein-ligand distance matrix and then optimizing the pose. In contrast, EquiBind [41] and E3Bind [51] directly predict the docked pose coordinates. Though efficient, the accuracy of these methods falls behind the sampling-based methods. Additionally, the variation in protein sizes often requires the use of external modules to first select suitable binding pockets, which can impact efficiency. Many of these methods rely on external modules to detect favorable binding sites in proteins. For example, TankBind [31] and E3Bind [51] take P2Rank [23] as priors for generating the pocket center candidates, which results in the need for a separate module (e.g., affinity prediction in TankBind and confidence module in E3Bind) for pocket selection, increasing the training and inference complexity.

To address these limitations, we propose FABind, a Fast and Accurate docking framework in an end-to-end way. FABind unifies pocket prediction and docking, streamlining the process within a single model architecture, which consists of a series of equivariant layers with geometry-aware updates, allowing for either pocket prediction or docking with only different configurations. Notably, for pocket prediction, we utilize lightweight configurations to maintain efficiency without sacrificing accuracy. In contrast to conventional pocket prediction, which only uses protein as input and forecasts multiple potential pockets, our method incorporates a specific ligand to pinpoint the unique pocket that the ligand binds to. Integrating the ligand into pocket prediction is crucial as it aligns with the fundamental characterization of the docking problem. In this way, we achieve a quick (without the need for external modules such as P2Rank) and precise pocket prediction (see Section 5.1).

Several strategies are additionally proposed in FABind to make a fast and accurate docking prediction. (1) Our pocket prediction module operates as the first layer in our model hierarchy and is jointly trained with the subsequent docking module to ensure a seamless, end-to-end process for protein-ligand docking prediction. We also incorporate a pocket center constraint using Gumbel-Softmax. This way assigns a probabilistic weighting to the inclusion of amino acids in the pocket, which helps to identify the most probable pocket center and improve the precision of the docking prediction. (2) We incorporate the predicted pocket into the docking module training using a scheduled sampling approach [4]. This way ensures consistency between the training and inference stages with respect to pocket leverage, thereby avoiding any mismatch that may arise from using the native pocket during training and the predicted pocket during inference. In this way, FABind is trained on a variety of possible pockets, allowing it to generalize well to new docking scenarios. (3) While directly predicting the ligand pose [41] and optimizing the coordinates based on the protein-ligand distance map [31] are both widely adopted in the sampling-based methods to ensure efficiency, we integrate both predictions in FABind to produce a more accurate pose prediction.

To evaluate the performance of our method, we conducted experiments on the binding structure prediction benchmark and compared our work with multiple existing methods. Our results demonstrate that FABind outperforms existing methods, achieving a mean ligand RMSD of $6.4$. This is a significant improvement over previous methods and demonstrates the effectiveness of our approach. Remarkably, our approach also demonstrates superior generalization ability, performing surprisingly well on unseen proteins. This suggests that our model can be applied to a wide range of docking scenarios and has the potential to be a useful tool in drug discovery. In addition to achieving superior performance, our method is also much more efficient during inference (e.g., $170\times$ faster than Diff-Dock), making it a fast and accurate binding framework. This efficiency is critical in real-world drug discovery scenarios, where time and resources are often limited.

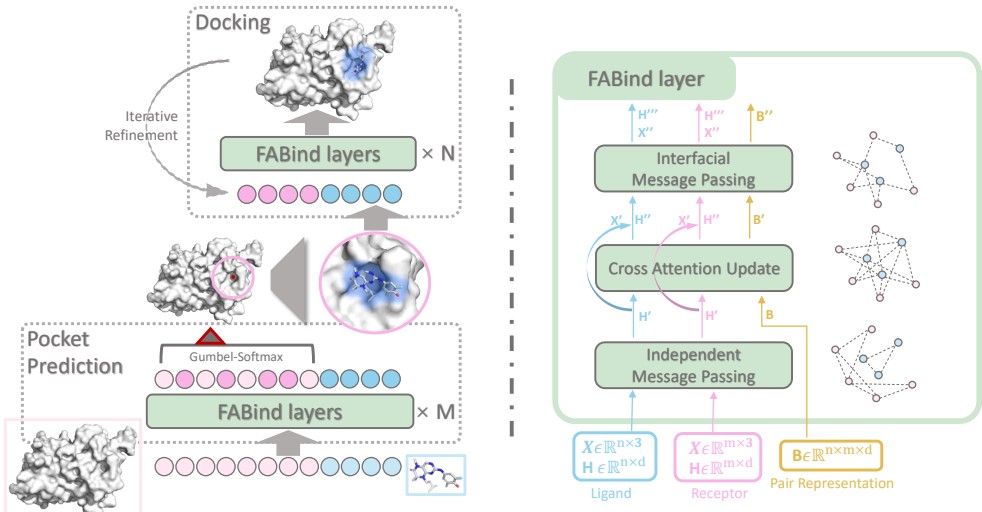

Figure 1: An overview of FABind. *Left*: The pocket prediction module takes the whole protein and the ligand as input and predicts the coordinates of the pocket center, where the ligand is randomly placed at the center of the protein. After determining the pocket center, a pocket is defined as a set of amino acids within a fixed radius around the center. Subsequently, the docking module moves the ligand to the pocket center and the ligand-pocket pair iteratively goes through the FABind layers to obtain the final pose prediction. $M$ and $N$ are the number of layers in pocket prediction and docking. *Right*: Architecture of FABind layers. Each layer contains three modules: independent message passing takes place within each component to update node embeddings and coordinates; cross-attention captures correlations between residues and ligands and updates embeddings only; and interfacial message passing focuses on the interface, attentively updating coordinates and representations.

## 2 Related Work

**Protein/Molecule Modeling.** Learning effective protein and molecule representations is fundamental to biology and chemistry research. Deep learning models have made great progress in protein and molecule presentation learning. Molecules are usually represented by graph or string formats, and the graph neural networks (e.g., GAT [46], GCN [48]) and sequence models (e.g., LSTM [12], Transformer [45]) are utilized to model the molecules. For proteins, the sequence models are the common choices since the FASTA sequence is the most widely adopted string format [38]. Considering the 3D representations, incorporating the geometric information/constraint, e.g., SE(3)-invariance or equivariance, is becoming promising to model the protein/molecule structures, and there are multiple equivariant/invariant graph neural networks proposed [39, 20, 3, 16]. Among them, AlphaFold2 [19] is almost the most successful geometry-aware modeling that achieves revolutionary performance in protein structure prediction. In our work, we also keep the equivariance when modeling the protein and ligand.

**Pocket Prediction.** Predicting binding pocket is critical in the early stages of structure-based drug discovery. Different approaches have been developed in the past years, including physical-chemical-based, geometric-based, and machine learning-based methods [40]. Typical geometric-based methods are SURFNET [26] and Fpocket [27], and many alternatives have been proposed [47, 6, 42]. Recently, machine learning-based, especially deep learning-based methods, have been promising, such as DeepSite [15] and DeepPocket [1] which directly predict the pocket using deep networks, and some works utilize the deep scoring functions (e.g., affinity score) to score and rank the pockets [36, 50]. Among them, P2Rank [23] is an open-sourced tool that is widely adopted in existing works [51, 31] to detect potential protein pockets. In this work, we distinguish ourselves from previous methods through the introduction of a novel equivariant module and the combination of two separate losses. Besides, we jointly train the pocket prediction module and the docking module, leveraging the knowledge gained from the docking module to improve the performance of pocket prediction.

**Protein-Ligand Docking.** Protein-ligand docking prediction is to predict the binding structure of the protein-ligand complex. Traditional methods usually take the physics-informed energy functions to score, rank, and refine the ligand structures, such as AutoDock Vina [43], SMINA [21], GLIDE [8]. Recently, geometric deep learning has been attractive and greatly advances docking prediction. There are two distinct approaches in docking research. Regression-based methods, such as EquiBind [41], TankBind [31], and E3Bind [51], directly predict the docked ligand pose. On the other hand, sampling-based methods, like DiffDock [7], require extensive ligand pose sampling and optimization, but often yield more accurate predictions. Our work lies in the research of regression-based methods with much less computational costs, but achieves comparable performance to sampling-based approaches.

## 3 Method

An overview of our proposed FABind is presented in Fig. 1. We first clarify the notations and problem definition in Section 3.1. In Section 3.2, we illustrate our FABind layer in detail. The specific design of the pocket prediction module is explained in Section 3.3, while the docking module is introduced in Section 3.4. Furthermore, the training pipeline is comprehensively described in Section 3.5.

### 3.1 Preliminaries

**Notations.** For each protein-ligand complex, we represent it as a graph and denote it as $\mathcal{G} = (\mathcal{V} := \{\mathcal{V}^l, \mathcal{V}^p\}, \mathcal{E} := \{\mathcal{E}^l, \mathcal{E}^p, \mathcal{E}^{lp}\})$. Specifically, the ligand subgraph is $\mathcal{G}^l = (\mathcal{V}^l, \mathcal{E}^l)$, where node $v_i = (\mathbf{h}_i, \mathbf{x}_i) \in \mathcal{V}^l$ is an atom, $\mathbf{h}_i$ is the pre-extracted feature by TorchDrug [52], and $\mathbf{x}_i \in \mathbb{R}^3$ is the corresponding coordinate. The number of atoms is denoted as $n^l$. $\mathcal{E}^l$ is the edge set that represents chemical bonds in the ligand. The protein subgraph is $\mathcal{G}^p = (\mathcal{V}^p, \mathcal{E}^p)$, where node $v_j = (\mathbf{h}_j, \mathbf{x}_j) \in \mathcal{V}^p$ is a residue, $\mathbf{h}_j$ is initialized with the pre-trained ESM-2 [28] feature following DiffDock [7], and $\mathbf{x}_j \in \mathbb{R}^3$ is the coordinate of the $C_\alpha$ atom in the residue. The number of residues is denoted as $n^p$. $\mathcal{E}^p$ is the edge set constructed by a cut-off distance 8Å. The input of the pocket prediction module is $\mathcal{G}$. We further denote the pocket subgraph as $\mathcal{G}^{p*} = (\mathcal{V}^{p*}, \mathcal{E}^{p*})$, with $n^{p*}$ residues in the pocket. The pocket and ligand form a new complex as the input to the docking module: $\mathcal{G}^{lp*} = (\mathcal{V} := \{\mathcal{V}^l, \mathcal{V}^{p*}\}, \mathcal{E} := \{\mathcal{E}^l, \mathcal{E}^{p*}, \mathcal{E}^{lp*}\})$, where $\mathcal{E}^{lp*}$ defines edges in the external contact surface. Detailed edge construction rule is in Appendix Section A.4. For clarity, we always use indices $i$, $k$ for ligand nodes, and $j$, $k'$ for protein nodes.

**Problem Definition.** Given a bounded protein and an unbounded ligand as inputs, our goal is to predict the binding pose of the ligand, denoted as $\{\mathbf{x}_i^{l*}\}_{1 \leq i \leq n_l}$. Following previous works, we focus on *blind docking* scenario in which we have zero knowledge about protein pocket.

### 3.2 FABind Layer

In this section, we provide a comprehensive description of the FABind layer. For clarity, we use FABind in the pocket prediction module for a demonstration.

**Overview.** Besides node-level information, we explicitly model pair embedding for each protein residue-ligand atom pair $(i, j)$. We follow E3bind [51] to construct a pair embedding $\mathbf{z}_{ij}$ via an outer product module (OPM): $\mathbf{z}_{ij} = \mathrm{Linear}\left((\mathrm{Linear}(\mathbf{h}_i) \bigotimes \mathrm{Linear}(\mathbf{h}_j))\right)$. For the initial pair embedding, OPM is operated on the transformed initial protein/ligand node embedding $\mathbf{h}_i/\mathbf{h}_j$. As illustrated in Figure 1, each FABind layer conducts three-step message passing: (1) Independent message passing. The independent encoder first passes messages inside the protein and ligand to update node embeddings and coordinates. (2) Cross-attention update. This block operates to exchange information across every node and updates pair embeddings accordingly. (3) Interfacial message passing. This layer focuses on the contact surface and attentively updates coordinates and representations for such nodes. The fundamental concept behind this design is the recognition of distinct characteristics between internal interactions within the ligand or protein and external interactions between the ligand and protein in biological functions. After several layers of alternations, we perform another independent message passing for further adjustment before the output of pocket prediction/docking modules. Notably, the independent and interfacial message passing layers are E(3)-equivariant, while cross-attention update layer is E(3)-invariant since it does not encode structure. These ensure each layer is E(3)-equivariant.

**Independent Message Passing.** We introduce a variant of Equivariant Graph Convolutional Layer (EGCL) proposed by EGNN [39] as our independent message passing layer. For simplicity, here we only illustrate the detailed message passing of ligand nodes, while the protein updates are in a similar way. With the ligand atom embedding $\mathbf{h}_i^l$ and the corresponding coordinate $\mathbf{x}_i^l$ in the $l$-th layer, we perform the independent message passing as follows:

$$\mathbf{m}_{ik} = \phi_e\left(\mathbf{h}_i^l, \mathbf{h}_k^l, \left\|\mathbf{x}_i^l - \mathbf{x}_k^l\right\|^2\right),$$

$$\mathbf{h}_i^l = \mathbf{h}_i^l + \phi_h\left(\mathbf{h}_i^l, \sum_{k\in\mathcal{N}(i|\mathcal{E}^l)}\mathbf{m}_{ik}\right), \quad \mathbf{x}_i^l = \mathbf{x}_i^l + \frac{1}{|\mathcal{N}(i|\mathcal{E}^l)|}\sum_{k\in\mathcal{N}(i|\mathcal{E}^l)}\left(\mathbf{x}_i^l - \mathbf{x}_k^l\right)\phi_x\left(\mathbf{m}_{ik}\right),$$

where $\phi_e, \phi_x, \phi_h$ are Multi-Layer Perceptons (MLPs) [10] and $\mathcal{N}(i|\mathcal{E}^l)$ denotes the neighbors of node $i$ regarding the internal edges $\mathcal{E}^l$ of the ligand.

**Cross-Attention Update.** After independent message passing, we enhance the node feature with cross-attention update over all protein/ligand nodes by passing the messages from all ligand/protein nodes. The pair embeddings are also updated accordingly. We also take the ligand embedding update for clarity. Given the ligand atom representations and the pair embeddings, we first perform multi-head cross-attention over all protein residues:

$$a_{ij}^{(h)} = \mathrm{softmax}_j\left(\frac{1}{\sqrt{c}}\mathbf{q}_i^{(h)^\top}\mathbf{k}_j^{(h)} + b_{ij}^{(h)}\right), \quad \mathbf{h}_i^l = \mathbf{h}_i^l + \mathrm{Linear}\left(\mathrm{concat}_{1\leq h\leq H}\left(\sum_{j=1}^{n^{p*}}a_{ij}^{(h)}\mathbf{v}_j^{(h)}\right)\right),$$

where $\mathbf{q}_i^{(h)}, \mathbf{k}_j^{(h)}, \mathbf{v}_j^{(h)}$ are linear projections of the node embedding, and $b_{ij}^{(h)} = \mathrm{Linear}(\mathbf{z}_{ij}^l)$ is a linear transformation of pair embedding $\mathbf{z}_{ij}^l$. The protein embeddings $\mathbf{h}_j^l$ are updated similarly. Based on updated node embeddings $\mathbf{h}_i^l$ and $\mathbf{h}_j^l$, the pair embeddings are further updated by $\mathbf{z}_{ij}^l = \mathbf{z}_{ij}^l + \mathrm{OPM}(\mathbf{h}_i^l, \mathbf{h}_j^l)$.

**Interfacial Message Passing.** With the updated protein and ligand representations, we perform an interfacial message passing to update the included node features and the coordinates on the contact surface. Our interfacial message passing derives from MEAN [22] with an additional attention bias. The detailed updates are as follows:

$$\alpha_{ij} = \frac{\exp\left(\mathbf{q}_i^\top\mathbf{k}_{ij} + b_{ij}\right)}{\sum_{j\in\mathcal{N}(i|\mathcal{E}^{lp*})}\exp\left(\mathbf{q}_i^\top\mathbf{k}_{ij} + b_{ij}\right)},$$

$$\mathbf{h}_i^{l+1} = \mathbf{h}_i^l + \sum_{j\in\mathcal{N}(i|\mathcal{E}^{lp*})}\alpha_{ij}\mathbf{v}_{ij}, \quad \mathbf{x}_i^{l+1} = \mathbf{x}_i^l + \sum_{j\in\mathcal{N}(i|\mathcal{E}^{lp*})}\alpha_{ij}\left(\mathbf{x}_i^l - \mathbf{x}_j^l\right)\phi_{xv}\left(\mathbf{v}_{ij}\right),$$

where $\mathbf{q}_i = \phi_q\left(\mathbf{h}_i^l\right)$, $\mathbf{k}_{ij} = \phi_k\left(\left\|\mathbf{x}_i^l - \mathbf{x}_j^l\right\|^2, \mathbf{h}_j^l\right)$, $\mathbf{v}_{ij} = \phi_v\left(\left\|\mathbf{x}_i^l - \mathbf{x}_j^l\right\|^2, \mathbf{h}_j^l\right)$, and $b_{ij} = \phi_b\left(\mathbf{z}_{ij}^l\right)$, and $\phi_q, \phi_k, \phi_v, \phi_b, \phi_{xv}$ are MLPs. $\mathcal{E}^{lp*}$ denotes the external edges between ligand and protein contact surface constructed by cut-off distance 10Å.

### 3.3 Pocket Prediction

In the pocket prediction module, given the protein-ligand complex graph $\mathcal{G}$, our objective is to determine the amino acids of the protein that belong to the pocket. Previous works such as TankBind and E3Bind both use P2Rank [23] to produce multiple pocket candidates. Subsequently, either affinity score (TankBind) or self-confidence score (E3Bind) is utilized to select the most appropriate docked pose. Though P2Rank is faster and better than previous tools, it is based on numerical algorithms and traditional machine learning classifiers. Furthermore, the incorporation of P2Rank necessitates the selection of candidate poses following multiple poses docking. These factors could potentially restrict the performance and efficiency of fully deep learning-based docking approaches.

In our work, we propose an alternative method by treating pocket prediction as a binary classification task on the residues using the FABind layer, where each residue in the protein is classified as belonging to the pocket or not. Hence, the pocket prediction is more unified with the deep learning docking. Specifically, we use a binary cross-entropy loss to train our pocket classifier:

$$\mathcal{L}_p^c = -\frac{1}{n_p}\sum_{j=1}^{n_p}[y_j\log(p_j) + (1 - y_j)\log(1 - p_j)],$$

where $n_p$ is the number of residues in the protein, and $y_j$ is the binary indicator for residue $j$ (i.e., 1 if it belongs to a pocket, 0 otherwise). $p_j = \sigma(\text{MLP}(\{\text{FABind layer}(\mathcal{G})\}_j))$ is the predicted probability of residue $j$ belonging to a pocket, $\mathcal{G}$ is the protein-ligand complex graph, and $\sigma$ is `sigmoid` function.

Besides the direct classification of each residue to decide the pocket, the common practice of leveraging P2Rank pocket prediction is to first predict a pocket center coordinate, then a sphere near the pocket center under a radius 20Å. Therefore, we add a constraint about the pocket center to make a more accurate prediction.

**Constraint for Pocket Center.** To constrain the predicted pocket center, we introduce a pocket center regression task over the classified pocket residues. Given $n^{p'}$ predicted pocket residues $\mathcal{V}^{p'} = \{v_1^{p'}, v_2^{p'}, ..., v_{n^{p'}}^{p'}\}$ from our classifier, the pocket center coordinate is $\mathbf{x}^{p'} = \frac{1}{n^{p'}} \sum_{j=1}^{n^{p'}} \mathbf{x}_j^{p'}$. Then we can add a distance loss between the predicted pocket center $\mathbf{x}^{p'}$ and the native pocket center $\mathbf{x}^{p*}$. This pocket center computation inherently involves discrete decisions – selecting which amino acids contribute to the pocket. Hence, we apply Gumbel-Softmax [14] to produce a differentiable approximation of the discrete selection process. It provides a probabilistic "hard" selection, which more accurately reflects the discrete decision to include or exclude an amino acid in the pocket.

$$\gamma_j^p = \frac{\exp((\log(p_j) + g_j)/\tau_e)}{\sum_{k'=1}^{n^p} \exp((\log(p_{k'}) + g_{k'})/\tau_e)},$$

where $g_j$ is sampled from Gumbel distribution $g_j = -\log(-\log U_m)$, $U_m \sim \text{Uniform}(0, 1)$, and $\tau_e$ is the controllable temperature. Then we use Huber loss [13] between the predicted pocket center $\mathbf{x}^p = \frac{1}{n^p} \sum_{j=1}^{n^p} \gamma_j^p \mathbf{x}_j^p$ and the native pocket center $\mathbf{x}^{p*}$ as the constraint loss,

$$\mathcal{L}_p^{c2r} = l_{Huber}(\mathbf{x}^p, \mathbf{x}^{p*}).$$

**Training Loss.** The pocket prediction loss is comprised of classification loss $\mathcal{L}_p^c$ and pocket center constraint loss $\mathcal{L}_p^{c2r}$, with a weight factor $\alpha = 0.2$,

$$\mathcal{L}_{pocket} = \mathcal{L}_p^c + \alpha \mathcal{L}_p^{c2r}.$$

**Pocket Decision.** Our pocket classifier can output the classified residues in a pocket. Following previous works [31, 51], we do not directly take these predicted residues as the pocket. Instead, we calculate the center of these classified residues and take it as the predicted pocket center, and the predicted pocket is in a sphere near the predicted center under a radius 20Å. However, for some proteins, our classification model may predict each residue as negative for a pocket, making it impossible to determine the pocket center. This is likely due to an imbalance between the pocket and non-pocket residues. For these rare cases, we take the Gumbel-softmax predicted center $x^p$ as the pocket center to address this problem.

### 3.4 Docking

In the docking task, given a pocket substructure $\mathcal{G}^{p*}$, our docking module predicts the coordinate of each atom in $\mathcal{G}^l$. The docking task is challenging since it requires the model to preserve E(3)-equivariance for every node while capturing pocket structure and chemical bonds of ligands.

**Iterative refinement.** Iterative refinement [19] is adopted in docking FABind layers to refine the structures by feeding the predicted ligand pose back to the message passing layers several rounds. During refinement iterations, new graphs are generated and the edges are also constructed dynamically.

After $k$ iterations (iterative refinement) of the $N\times$ FABind layer alternations, we obtain the final coordinates $\mathbf{x}^L$ and node embeddings $\mathbf{h}_i^L$ and $\mathbf{h}_j^L$. Besides directly optimizing the coordinate loss, we additionally add distance map constraints to refine the ligand pose better. We reconstruct distance matrices in two ways. One is to directly compute based on the predicted coordinates: $\widetilde{D}_{ij} = \|\mathbf{x}_i^L - \mathbf{x}_j^L\|$. The other is to predict from the pair embeddings $\mathbf{z}_{ij}^L$ by an MLP transition: $\widehat{D}_{ij} = \text{MLP}(\mathbf{z}_{ij}^L)$, where each vector outputs a distance scalar.

**Training Loss.** The docking loss is comprised of coordinate loss $\mathcal{L}_{coord}$ and distance map loss $\mathcal{L}_{dist}$:

$$\mathcal{L}_{docking} = \mathcal{L}_{coord} + \beta \mathcal{L}_{dist}.$$

| | **Ligand RMSD** | | | | | | **Centroid Distance** | | | | | | **Average** |
|---|---|---|---|---|---|---|---|---|---|---|---|---|---|
| | Percentiles ↓ | | | | % Below ↑ | | Percentiles ↓ | | | | % Below ↑ | | |
| **Methods** | 25% | 50% | 75% | Mean | 2Å | 5Å | 25% | 50% | 75% | Mean | 2Å | 5Å | **Runtime (s)** |
| QVina-W | 2.5 | 7.7 | 23.7 | 13.6 | 20.9 | 40.2 | 0.9 | 3.7 | 22.9 | 11.9 | 41.0 | 54.6 | 49* |
| GNINA | 2.8 | 8.7 | 22.1 | 13.3 | 21.2 | 37.1 | 1.0 | 4.5 | 21.2 | 11.5 | 36.0 | 52.0 | 146 |
| SMINA | 3.8 | 8.1 | 17.9 | 12.1 | 13.5 | 33.9 | 1.3 | 3.7 | 16.2 | 9.8 | 38.0 | 55.9 | 146* |
| GLIDE | 2.6 | 9.3 | 28.1 | 16.2 | 21.8 | 33.6 | 0.8 | 5.6 | 26.9 | 14.4 | 36.1 | 48.7 | 1405* |
| VINA | 5.7 | 10.7 | 21.4 | 14.7 | 5.5 | 21.2 | 1.9 | 6.2 | 20.1 | 12.1 | 26.5 | 47.1 | 205* |
| EQUIBIND | 3.8 | 6.2 | 10.3 | 8.2 | 5.5 | 39.1 | 1.3 | 2.6 | 7.4 | 5.6 | 40.0 | 67.5 | **0.03** |
| TANKBIND | 2.6 | 4.2 | 7.6 | 7.8 | 17.6 | 57.8 | 0.8 | 1.7 | 4.3 | 5.9 | 55.0 | 77.8 | 0.87 |
| E3BIND | 2.1 | 3.8 | 7.8 | 7.2 | 23.4 | 60.0 | 0.8 | 1.5 | 4.0 | 5.1 | 60.0 | 78.8 | 0.44 |
| DIFFDOCK (1) | 2.4 | 4.9 | 8.9 | 8.3 | 20.4 | 51.0 | 0.7 | 1.8 | 4.5 | 5.8 | 54.1 | 76.8 | 2.72 |
| DIFFDOCK (10) | 1.6 | 3.8 | 7.9 | 7.4 | 32.4 | 59.7 | 0.6 | 1.4 | 3.6 | 5.2 | 60.7 | 79.8 | 20.81 |
| DIFFDOCK (40) | **1.5** | 3.5 | 7.4 | 7.4 | **36.0** | 61.7 | **0.5** | **1.2** | **3.3** | 5.4 | **62.9** | **80.2** | 82.83 |
| FABIND | 1.7 | **3.1** | **6.7** | **6.4** | 33.1 | **64.2** | 0.7 | 1.3 | 3.6 | **4.7** | 60.3 | **80.2** | 0.12 |

Table 1: Flexible blind self-docking performance. The top half contains results from traditional docking software; the bottom half contains results from recent deep learning based docking methods. The last line shows the results of our FABind. The number of poses that DiffDock samples is specified in parentheses. We run the experiments of DiffDock three times with different random seeds and report the mean result for robust comparison. The symbol "*" means that the method operates exclusively on the CPU. The superior results are emphasized by bold formatting, while those of the second-best are denoted by an underline.

$\mathcal{L}_{coord}$ is computed as the Huber distance between the predicted coordinates and ground truth coordinates of the ligand atoms. $\mathcal{L}_{dist}$ is comprised of three terms, each of which is $\mathcal{L}_2$ loss between different components of the ground truth and the two reconstructed distance maps. Formally,

$$\mathcal{L}_{dist} = \frac{1}{n^l n^{p*}} [\sum_{i=1}^{n^l} \sum_{j=1}^{n^{p*}} (D_{ij} - \widetilde{D}_{ij})^2 + \sum_{i=1}^{n^l} \sum_{j=1}^{n^{p*}} (D_{ij} - \widehat{D}_{ij})^2 + \gamma \sum_{i=1}^{n^l} \sum_{j=1}^{n^{p*}} (\widetilde{D}_{ij} - \widehat{D}_{ij})^2],$$

where $D_{ij}$ is ground truth distance matrix. In practice, we set $\beta = \gamma = 1.0$.

### 3.5 Pipeline

We first predict the protein pocket and then predict the docked ligand coordinates in a hierarchical unified framework. However, the common approach is to only use the native pocket for docking in the training phase, which is known as teacher-forcing [24] training. Therefore, there is a mismatch in that the training phase takes the native pocket while the inference phase can only take the predicted pocket since we do not know the native pocket in inference. To reduce this gap, we incorporate a scheduled training strategy to gradually involve the predicted pocket in the training stage instead of using the native pocket only. Specifically, our training pipeline consists of two stages, (1) in the initial stage, since the performance of pocket prediction is poor, we only use the native pocket to perform the docking training; (2) In the second stage, with the improved pocket prediction ability, we then involve the predicted pocket into docking, where the native pocket is still kept in docking. The ratio between the predicted pocket and the native pocket is $1 : 3$.

**Comprehensive Training Loss.** Our comprehensive training loss comprises two components: the pocket prediction loss and the docking loss,

$$\mathcal{L} = \mathcal{L}_{pocket} + \mathcal{L}_{docking}. \tag{1}$$

## 4 Experiments

### 4.1 Setting

**Dataset.** We conduct experiments on the PDBbind v2020 dataset [29], which is from Protein Data Bank (PDB) [5] and contains 19,443 protein-ligand complex structures. To maintain consistency with prior works [41, 31], we follow similar preprocessing steps (see Appendix Section A.1 for more details). After the filtration, we used $17,299$ complexes that were recorded before 2019 for training purposes, and an additional 968 complexes from the same period for validation. For our testing phase, we utilized 363 complexes recorded after 2019.

| Methods | Ligand RMSD | | | | | | Centroid Distance | | | | | | Average |
| | Percentiles ↓ | | | | % Below ↑ | | Percentiles ↓ | | | | % Below ↑ | | |
| | 25% | 50% | 75% | Mean | 2Å | 5Å | 25% | 50% | 75% | Mean | 2Å | 5Å | Runtime (s) |
|---|---|---|---|---|---|---|---|---|---|---|---|---|---|
| QVINA-W | 3.4 | 10.3 | 28.1 | 16.9 | 15.3 | 31.9 | 1.3 | 6.5 | 26.8 | 15.2 | 35.4 | 47.9 | 49* |
| GNINA | 4.5 | 13.4 | 27.8 | 16.7 | 13.9 | 27.8 | 2.0 | 10.1 | 27.0 | 15.1 | 25.7 | 39.5 | 146 |
| SMINA | 4.8 | 10.9 | 26.0 | 15.7 | 9.0 | 25.7 | 1.6 | 6.5 | 25.7 | 13.6 | 29.9 | 41.7 | 146* |
| GLIDE | 3.4 | 18.0 | 31.4 | 19.6 | **19.6** | 28.7 | 1.1 | 17.6 | 29.1 | 18.1 | 29.4 | 40.6 | 1405* |
| VINA | 7.9 | 16.6 | 27.1 | 18.7 | 1.4 | 12.0 | 2.4 | 15.7 | 26.2 | 16.1 | 20.4 | 37.3 | 205* |
| EQUIBIND | 5.9 | 9.1 | 14.3 | 11.3 | 0.7 | 18.8 | 2.6 | 6.3 | 12.9 | 8.9 | 16.7 | 43.8 | **0.03** |
| TANKBIND | 3.4 | 5.7 | 10.8 | 10.5 | 3.5 | 43.7 | 1.2 | 2.6 | 8.4 | 8.2 | 40.9 | 70.8 | 0.87 |
| E3BIND | 3.0 | 6.1 | 10.2 | 10.1 | 6.3 | 38.9 | 1.2 | 2.3 | 7.0 | 7.6 | 43.8 | 66.0 | 0.44 |
| DIFFDOCK (1) | 4.1 | 7.2 | 18.2 | 12.5 | 8.1 | 33.1 | 1.4 | 3.7 | 16.7 | 10.0 | 33.6 | 58.3 | 2.72 |
| DIFFDOCK (10) | 3.2 | 6.4 | 16.5 | 11.8 | 14.2 | 38.7 | 1.1 | 2.8 | 13.3 | 9.3 | 39.7 | 62.6 | 20.81 |
| DIFFDOCK (40) | 2.8 | 6.4 | 16.3 | 12.0 | 17.2 | 42.3 | 1.0 | 2.7 | 14.2 | 9.8 | 43.3 | 62.6 | 82.83 |
| FABIND | **2.2** | **3.4** | **8.3** | **7.7** | 19.4 | **60.4** | **0.9** | **1.5** | **4.7** | **5.9** | **57.6** | **75.7** | 0.12 |

Table 2: Flexible blind self-docking performance on unseen receptors.

**Model Configuration.** For pocket prediction, we use $M = 1$ FABind layer with 128 hidden dimensions. The pocket radius is set to 20Å. As for docking, we use $N = 4$ FABind layers with 512 hidden dimensions. Each layer comprises three-step message passing modules (refer to Fig. 1). We perform a total of $k = 8$ iterations for structure refinement during the docking process. Additional training details can be found in the Appendix Section A.2.

**Evaluation.** Following EquiBind [41], the evaluation metrics are (1) Ligand RMSD, which calculates the root-mean-square deviation between predicted and true ligand atomic Cartesian coordinates, indicating the model capability of finding the right ligand conformation at atom level; (2) Centroid Distance, which calculates the Euclidean distance between predicted and true averaged ligand coordinates, reflecting the model capacity to identify the right binding site. To generate an initial ligand conformation, we employed the ETKDG algorithm [37] using RDKit [25], which randomly produces a low-energy ligand conformation.

## 4.2 Performance in Flexible Self-docking

In the context of flexible blind self-docking, where the bound ligand conformation is unknown and the model is tasked with predicting both the bound ligand conformation and its translation and orientation, we observe the notable performance of our FABind. Across most metrics, FABind ranking either as the best or second best, as presented in Table 1. Our FABind exceeds the DiffDock with 10 sampled poses. Although FABind may not achieve the best performance in terms of the <2Å metric, it demonstrates exceptional proficiency with a mean RMSD score of 6.4Å. The comparatively lower performance in the <2Å metric can be attributed to the optimization objective of DiffDock, which primarily focuses on optimizing ligand poses towards achieving <2Å accuracy. FABind also gets comparative performance on Centroid Distance metrics. Overall, the experimental results showcase the remarkable efficacy of FABind.

## 4.3 Performance in Self-docking for Unseen Protein

To perform a more rigorous evaluation, we apply an additional filtering step to exclude samples whose UniProt IDs of the proteins are not contained in the data that is seen during training and validation. The resulting subset, consisting of 144 complexes, was then used to evaluate the performance of FABind. The results of this evaluation are presented in Table 2, demonstrating that FABind surpasses other deep learning and traditional methods by a significant margin across all evaluation metrics. These findings strongly indicate the robust generalization capability of FABind.

## 4.4 Inference Efficiency

We conducted an inference study to demonstrate the efficiency of FABind. The statistics regarding the average time cost per sample are presented in Table 1. Comparing our method to others, we observe that the inference time cost of FABind ranks second lowest. Notably, when compared to DiffDock with 10 sampled poses, our FABind achieves better docking performance while being over 170 times faster. The efficiency of our method can be attributed to eliminating the need for an external

pocket selection module, and the sampling and scoring steps used by other methods. By doing so, our FABind produces significantly faster docking. While EquiBind exhibits superior inference speed, it performs poorly in the docking task. These compelling findings strongly support the claim that our FABind is not only highly fast but also accurate.

## 5 Further Analysis

### 5.1 Pocket Prediction Analysis

In this study, We conduct an independent analysis of our pocket prediction module. To evaluate its performance, we compared it with two existing methods, TankBind and E3Bind, both of which use P2Rank to first segment the protein to multiple pocket blocks, and then apply a confidence module for ranking and selection. The selected pockets are used to compare with our predicted pockets. The pocket prediction performance is measured by the distance between the predicted pocket center and the center of the native binding site (DCC) metric, which is a widely-used metric in pocket prediction task [1]. We use thresholds of 3Å, 4Å, and 5Å, to determine successful predictions. From Table 3, we can see that: (1) removing the center constraint (classification only) significantly hurts the prediction performance. (2) The integration of ligand information into the pocket prediction task also enhances performance and efficiency. Previous two-stage methods involve initial docking for multiple pocket candidates, followed by the selection of the optimal candidate. Nevertheless, the majority of potential binding sites are irrelevant to the particular ligand [51]. Our FABind further improves the inference efficiency as our pocket prediction is essentially designed for predicting the most probable pocket for the specific ligand. (3) Combined with center constraint, even without incorporating ligand information, our pocket prediction performance outperforms TankBind and E3Bind, which shows our rational design of the pocket prediction module is effective.

|  | DCC % Below ↑ | | |
| --- | --- | --- | --- |
| **Methods** | 3Å | 4Å | 5Å |
| TANKBIND | 18.2 | 32.0 | 39.9 |
| E3BIND | 26.7 | 35.8 | 50.1 |
| P2RANK | 36.4 | 50.1 | 57.0 |
| FABIND | **42.7** | **56.5** | **62.8** |
| - LIGAND INFORMATION | 36.9 | 51.5 | 59.0 |
| - CENTER CONSTRAINT | 8.8 | 22.9 | 31.7 |

Table 3: Results of pocket prediction.

### 5.2 Ablation Study

In this subsection, we perform several ablation studies to analyze the contribution of different components, the results for selected metrics are reported in Table 4. We examine four specific settings: removal of the scheduled training strategy (teacher forcing); removal of the distance map related losses; removal of the iterative refinement; and removal of the cross-attention update. From the table, we can see that: (1) each of the components contributes to the good performance of our FABind. (2) The combination of distance map losses impacts most of the docking performance. (3) The remaining components primarily contribute to improvements in fine-grained scales (<2Å). A full ablation study can be found in Appendix Section C.1.

| Methods | RMSD Mean (Å)↓ | RMSD % Below 2Å↑ |
| --- | --- | --- |
| FABIND | 6.4 | 33.1 |
| NO SCHEDULED SAMPLING | 6.4 | 28.7 |
| COORD LOSS ONLY | 6.9 | 16.3 |
| NO ITERATIVE REFINEMENT | 6.6 | 22.5 |
| NO CROSS-ATTENTION | 6.4 | 21.4 |

Table 4: Results of ablation study.

### 5.3 Case Study

**FABind can identify the right binding pocket for a new large protein.** In Fig. 2(a), the large protein target (PDB 6NPI) is not seen during training, thus posing a significant challenge to accurately locate the binding pocket. Though predicted poses of other deep learning models (DiffDock, E3Bind, TankBind, and EquiBind) are all off-site, FABind can successfully identify the native binding site and predict the binding pose (in green) with low RMSD (3.9Å), showing the strong generalization ability.

**FABind generates ligand pose with better performance and validity.** Fig. 2(b) shows the docked results of another protein (PDB 6G3C). All methods find the right pocket, but our FABind aligns best RMSD (0.8Å) with the ground truth ligand pose. In comparison, E3Bind produces a knotted pose where rings are knotted together which is not valid. DiffDock also produces the same accurate ligand

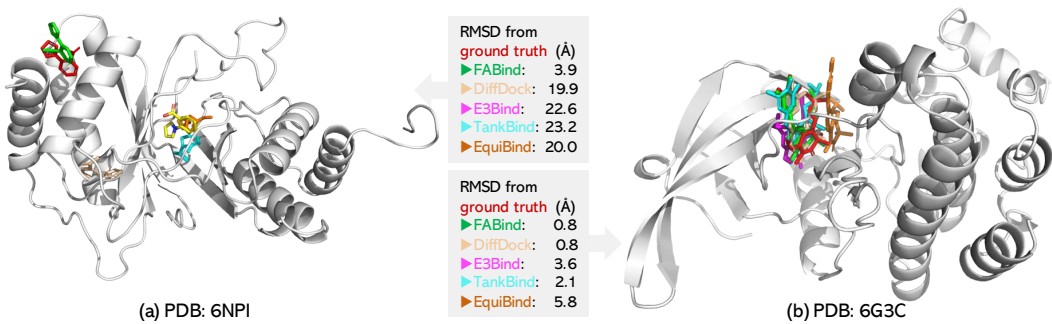

Figure 2: Case studies. Pose prediction by FABind (green), DiffDock (wheat), E3Bind (magenta), TankBind (cyan), and EquiBind (orange) are placed together with protein target, and RMSDs to ground truth (red) are reported. (a) For large unseen protein (PDB 6NPI), FABind successfully identifies the pocket, while the others are all off-site. (b) For the other protein (PDB 6G3C), all models find the right pocket, among which FABind predicts the most precise and valid binding pose as the DiffDock but with faster speed.

pose but is much slower. These show that FABind can not only find good pose with high speed but also maintain structural rationality.

# 6 Conclusion

In this paper, we propose an end-to-end framework, FABind, with several rational strategies. We propose a pocket prediction mechanism to be jointly trained with the docking module and bridge the gap between pocket training and inference with a scheduled training strategy. We also introduce a novel equivariant layer for both pocket prediction and docking. For ligand pose prediction, we incorporate both direct coordinate optimization and the protein-ligand distance map-based refinement. Empirical experiments show that FABind outperforms most existing methods with higher efficiency. In future work, we aim to better align the pocket prediction and docking modules and develop more efficient and effective tools.

# 7 Acknowledgments

We would like to thank the reviewers for their insightful comments. This work was partially supported by National Natural Science Foundation of China (NSFC Grant No. 62122089), Beijing Outstanding Young Scientist Program NO. BJJWZYJH012019100020098, and Intelligent Social Governance Platform, Major Innovation & Planning Interdisciplinary Platform for the "Double-First Class" Initiative, Renmin University of China.

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

# A  More Detailed Descriptions

## A.1  Dataset Preprocessing

As we stated in paper Section 4.1, PDBBind v2020 dataset [29] contains 19,443 ligand-protein complex structures, and we pre-process the structures as follows. First, we only keep complex structures whose ligand structure file (in `sdf` or `mol2` format) can be processed by RDKit [25] or TorchDrug [52], leaving 19,126 complexes. Then, to address the multiple equally valid binding ligand pose issue for symmetric receptor structures, we only keep the protein chains that have an atom within 10 Å radius of any atom of the ligand. We further filter out complexes in which the contact (distance is within 10Å) number between ligand atom and protein amino acid $C_\alpha$ are less than or equal to 5, or the number of ligand atom is more than or equal to 100. After applying these filters, $18,630$ complexes are left. Finally, we process the remaining complexes with the time split as described in EquiBind [41].

## A.2  Experiment Settings

**Baseline.** We compare our model with traditional score-based docking methods and recent geometry-based deep learning methods. For traditional docking methods, QVina-W, GNINA [33], SMINA [21], GLIDE [8] and AutoDock Vina [43] are included. For deep learning methods, EquiBind [41], TankBind [31], E3Bind [51] and DiffDock [7] are included.

We report corrected results for the deep learning baselines including EquiBind, TankBind, and E3Bind. The corrected results adopt post-optimization methods on model outputs, including fast point cloud fitting (used in EquiBind) and gradient descent (used in TankBind and E3Bind), which can further enforce geometry constraints within the ligand. For TankBind, the post-optimization method is used to get final ligand coordinates through the predicted distance matrix, which is essential for distance to coordinate transformation. However, for a fair comparison, the reported average runtime of EquiBind, TankBind, E3Bind, and FABind is the uncorrected version without post-optimization. The reported baseline results are mainly derived from the original paper of E3Bind [51]. For the DiffDock results, as the results of the sample-based method are unstable, we reproduce the results using its pre-trained checkpoint[4]. Specifically, we run the DiffDock inference codes with three random seeds and report the average results.

**Training and Evaluation.** The training process consists of two stages. In the initial warm-up stage, only the native pockets are used for docking. Once the pocket prediction performance reaches a certain threshold (specifically, when the center coordinate distance between the predicted center and ground truth is less than 4Å), the training progresses to the second stage. During the second stage, the predicted pockets are integrated into the docking training process. The sampling protocol involves a 75% probability of selecting the ground truth pocket and a 25% probability of selecting the predicted pocket. Note that the task of pocket prediction is consistently incorporated into the entire training process. Following E3Bind [51], we also apply normalization (divided by 5) and unnormalization (multiplied by 5) techniques on the coordinate and distance. Additionally, to improve the generalization ability of the model, the pocket is randomly shifted from $-5$Å to 5Å in all three axes during training. FABind models are trained for approximately 500 epochs using the AdamW [30] optimizer on 8 NVIDIA V100 16GB GPU with batch size set to 3 on each GPU. The learning rate is $5e-5$, which is scheduled to warm up in the first 15 epochs and then decay linearly. To further enforce geometric constraints, we also incorporate local atomic structures (LAS) constraints in the training process by ensuring the distances between ligand atoms $i$ and $k$ in the transformed conformer ($\mathbf{X}$) by the model are consistent with those in the initial low-energy conformer ($\mathbf{Z}$) for atoms that are either 2-hop neighbors or in the same ring structure, as proposed in EquiBind [41].

## A.3  Ligand and Protein Feature Encoding

As we stated in paper Section 3.1, we construct ligand feature by TorchDrug [52] toolkit and protein feature with the pre-trained ESM-2 model. Here we give a detailed description of the encoding. For ligand compound, the node embedding $\mathbf{h}_i$ is a 56-dimensional vector containing the following features: atom number; degree; the number of connected hydrogens; total valence; formal charge;

---

[4]`https://github.com/gcorso/DiffDock`

Table 5: **PDBBind blind docking on apo proteins.** The top half contains results from traditional docking software; the bottom half contains results from recent deep learning based docking methods. The last line shows the results of our FABind. No method received further training on ESMFold generated structures.

| Method | Apo ESMFold proteins Top-1 RMSD | |
|---|---|---|
| | %<2 | Med. |
| GNINA | 2.0 | 22.3 |
| SMINA | 3.4 | 15.4 |
| EQUIBIND | 1.7 | 7.1 |
| TANKBIND | 10.4 | 5.4 |
| P2RANK+SMINA | 4.6 | 10.0 |
| P2RANK+GNINA | 8.6 | 11.2 |
| EQUIBIND+SMINA | 4.3 | 8.3 |
| EQUIBIND+GNINA | 10.2 | 8.8 |
| DIFFDOCK (10) | 21.7 | 5.0 |
| DIFFDOCK (40) | 20.3 | 5.1 |
| FABIND | **24.9** | **4.2** |

whether or not it is in an aromatic ring. For protein target, we directly use the pre-trained 33-layer ESM-2 [28] model[5], which contains 650M parameters and is trained on UniRef 50M dataset. The node feature in the protein graph is derived from the amino acid feature and the hidden size is 1280.

## A.4 Model Architecture Details

**Edge Construction.** We now introduce how to construct the edges in our FABind layers. For clarity, we use FABind in the pocket prediction module for a demonstration. The cut-offs are set to the same in both pocket prediction and docking. As defined in the paper, we have three types of edges, $\mathcal{E} := \{\mathcal{E}^l, \mathcal{E}^p, \mathcal{E}^{lp}\}$, for ligand, protein, and ligand-protein interface, respectively. $\mathcal{E}^l$ and $\mathcal{E}^p$ are constructed from independent context, while $\mathcal{E}^{lp}$ is constructed from external interface. For the independent context of a ligand, we directly refer to chemical bonds as constructed edges $\mathcal{E}^l$ with the biological insight that a molecule keeps its chemical bonds during the process. For the independent context of a protein, $\mathcal{E}^p$ is defined as the edges connecting to nodes when the spatial distance is below a cutoff distance $c_{in}$. We set $c_{in} = 8.0$ in our work following Kong et al. [22]. Note that independent edges for ligands and proteins differ. For external edges $\mathcal{E}^{lp}$, we also add edges with a spatial radius threshold $c_{ex} = 10.0$ following TankBind [31].

**Global Nodes.** MEAN [22] demonstrates that the global nodes intensify information exchange during message passing. Therefore, we insert a global node into each component (ligand or protein) of the complex as well. A global node connects to all nodes in the same component and the other global node. The coordinates are initialized as zero tensors and can be updated during feed-forward.

**Iterative Refinement.** Iterative structure refinement has been proved as a critical design in structure prediction task [19]. It allows the network to go deeper without adding much computational overhead. Specifically, we update coordinates during all iterations and update hidden representations only in the last iteration. To stabilize the training process and save memories, we stop the gradients except for the last iteration. In our implementation, we also accelerate training speed by randomly sampling an iteration number less than or equal to the configuration $N$ for each batch, while always refining $N$ iterations during inference.

---

[5]The pre-trained ESM-2 checkpoint can be found at `https://dl.fbaipublicfiles.com/fair-esm/models/esm2_t33_650M_UR50D.pt`

| Methods | Ligand RMSD | | | | | | Centroid Distance | | | | | |
|---|---|---|---|---|---|---|---|---|---|---|---|---|
| | Percentiles ↓ | | | | % Below ↑ | | Percentiles ↓ | | | | % Below ↑ | |
| | 25% | 50% | 75% | Mean | 2Å | 5Å | 25% | 50% | 75% | Mean | 2Å | 5Å |
| FABIND | 1.7 | 3.1 | 6.7 | 6.4 | 33.1 | 64.2 | 0.7 | 1.3 | 3.6 | 4.7 | 60.3 | 80.2 |
| NO SCHEDULED SAMPLING | 1.7 | 3.5 | 7.0 | 6.4 | 28.7 | 63.4 | 0.7 | 1.5 | 3.5 | 4.5 | 60.3 | 82.6 |
| COORD LOSS ONLY | 2.6 | 4.1 | 7.3 | 6.9 | 16.3 | 60.9 | 1.0 | 1.7 | 4.5 | 4.7 | 53.9 | 77.4 |
| COORD LOSS + DIST. LOSS (1) | 1.9 | 3.5 | 7.4 | 6.5 | 26.2 | 63.4 | 0.7 | 1.5 | 4.1 | 4.5 | 56.7 | 80.0 |
| NO CROSS-ATTENTION | 2.2 | 4.2 | 7.0 | 6.4 | 21.4 | 59.9 | 0.9 | 1.9 | 4.1 | 4.5 | 52.3 | 80.4 |
| NO INDEPENDENT MP | 2.1 | 3.8 | 7.0 | 6.6 | 22.6 | 61.7 | 0.8 | 1.5 | 3.3 | 4.6 | 59.8 | 82.6 |
| NO ITERATIVE | 2.2 | 4.1 | 7.2 | 6.6 | 22.5 | 58.5 | 0.9 | 1.9 | 4.7 | 4.7 | 52.3 | 80.4 |
| NO ESM-2 | 1.9 | 3.6 | 8.0 | 6.3 | 27.5 | 61.2 | 0.8 | 1.6 | 3.8 | 4.4 | 56.2 | 79.1 |
| WITH POST OPTIMIZATION | 1.8 | 3.5 | 7.2 | 6.6 | 30.9 | 61.4 | 0.7 | 1.4 | 3.5 | 4.7 | 59.8 | 80.7 |

Table 6: Results of full ablation study.

## B Apo-Structure Docking

As stated in DiffDock [7], the PDBBind benchmark primarily focuses on evaluating the ability of docking methods to dock ligand to its corresponding receptor holo-structure, which is a simplified and less realistic scenario. However, in real-world applications, docking is often performed on apo or holo-structures that are bound to different ligands. To address this limitation, DiffDock proposed a new benchmark that combines the crystal complex structures of PDBBind with protein structures predicted by ESMFold [28]. In order to validate the efficacy of our FABind in the apo-structure docking scenario, we also evaluated its performance under the same settings with DiffDock. In order to validate the efficacy of our FABind in the apo-structure docking scenario, we implement the same experimental setup as DiffDock. For 363 test samples, we first extract their sequences from PDBBind and use `esmfold_v1` to predict their structures, from which process we assume that the apo protein structures are obtained. One sample (PDB: 6OTT) is filtered out due to out-of-memory error. Then we align the rest of the samples with PDBBind, where 12 samples are further excluded due to memory limitations, the same as DiffDock. The results in Table 5 demonstrate that FABind outperforms DiffDock, achieving an RMSD of less than 2Å on 24.9% of the complexes generated by ESMFold. This demonstrates the strong predictive capacity of FABind for apo-structure predictions.

## C Study

### C.1 Full Ablation

The comprehensive ablations are listed in Table 6. We can observe that each of the components contributes to the good performance of our FABind. Firstly, the scheduled training strategy, when removed, leads to a slight decrease in performance for challenging cases (e.g., RMSD 75%). This indicates that the scheduled training strategy contributes positively to handling difficult scenarios. Regarding loss construction, the inclusion of the distance map loss is crucial, and solely utilizing the first term of distance map loss (i.e., the distance loss between $D_{ij}$ and $\widetilde{D}_{ij}$ in $\mathcal{L}_{dist}$, which we call "coord loss + dist. loss (1)" in Table 6) does not yield optimal results. The architecture design also has a substantial impact. The removal of the cross-attention update and independent message passing severely impair the model's ability to handle favorable cases (e.g., RMSD 25%). This suggests that cross-attention update and independent message passing are both vital for capturing important structural dependencies. Furthermore, iterative refinement is found to be indispensable for most structure prediction models, including ours. In terms of feature representation, the utilization of ESM-2 features for residues proves to be most beneficial for challenging cases. It enhances the model's capability to handle difficult scenarios effectively. Lastly, post-optimization does not significantly affect the overall performance but ensures that the generated ligand conformations are chemically more rational.

### C.2 Iterative Refinement

In this section, we investigate the impact of iterative refinement on our model. We utilize our best-performing model and evaluate its performance using different iterations during inference. The

| | Ligand RMSD | | | | | | Centroid Distance | | | | | | Average |
| Iteration | Percentiles ↓ | | | | % Below ↑ | | Percentiles ↓ | | | | % Below ↑ | | |
| | 25% | 50% | 75% | Mean | 2Å | 5Å | 25% | 50% | 75% | Mean | 2Å | 5Å | Runtime (s) |
|---|---|---|---|---|---|---|---|---|---|---|---|---|---|
| ITERATION=1 | 3.1 | 4.8 | 7.9 | 7.3 | 5.2 | 51.8 | 1.2 | 2.0 | 5.3 | 5.3 | 49.9 | 74.1 | 0.03 |
| ITERATION=2 | 2.4 | 3.8 | 7.7 | 6.8 | 20.1 | 60.1 | 0.9 | 1.5 | 4.6 | 4.9 | 56.5 | 77.1 | 0.05 |
| ITERATION=4 | 1.8 | 3.3 | 7.2 | 6.5 | 28.4 | 63.6 | 0.7 | 1.4 | 3.8 | 4.8 | 59.5 | 78.8 | 0.07 |
| ITERATION=6 | 1.8 | 3.2 | 7.0 | 6.4 | 30.9 | 64.5 | 0.7 | 1.3 | 3.7 | 4.7 | 60.6 | 79.6 | 0.10 |
| ITERATION=8 | 1.7 | 3.1 | 6.7 | 6.4 | 33.1 | 64.2 | 0.7 | 1.3 | 3.6 | 4.7 | 60.3 | 80.2 | 0.12 |
| ITERATION=10 | 1.7 | 3.1 | 6.6 | 6.4 | 32.5 | 65.3 | 0.7 | 1.3 | 3.5 | 4.7 | 60.6 | 80.4 | 0.16 |
| ITERATION=12 | 1.7 | 3.1 | 6.6 | 6.4 | 32.5 | 65.6 | 0.7 | 1.4 | 3.6 | 4.7 | 61.7 | 80.4 | 0.19 |

Table 7: Inference on different iterations.

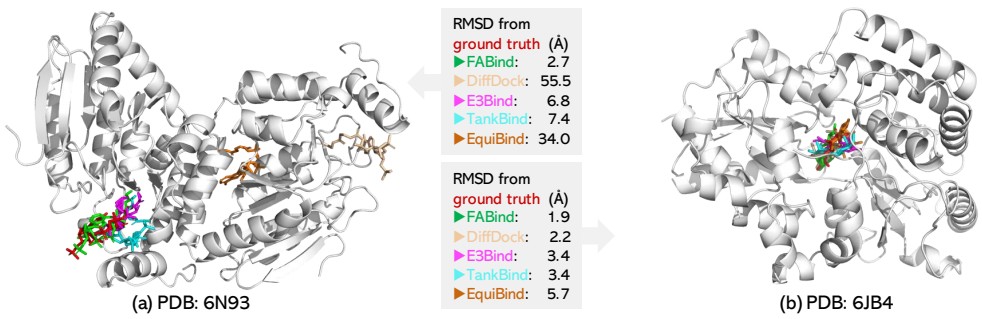

Figure 3: Additional case studies. Pose prediction by FABind (green), DiffDock (wheat), E3Bind (magenta), TankBind (cyan), and EquiBind (orange) are placed together with protein target structure, and RMSD to ground truth (red) are reported. (a) For unseen protein (PDB 6N93), FABind, E3Bind, and TankBind successfully identify the pocket, among which FABind predicts the most precise binding pose with the lowest RMSD 2.7Å, while the other methods are all off-site. (b) For PDB 6JB4, all deep learning models find the right pocket, among which FABind predicts the most precise binding pose with the lowest RMSD 1.9Å.

results are reported in Table 7. From the table, we can find that (denote the number of iterations as $i$, $1 \leq i \leq 12$): (1) the performance improves as $i$ increases. (2) The results tend to be stable when $i$ increases to some extent. (3) When $i = 8$, the results are generally optimal. However, the results are similar when $4 \leq i \leq 12$. Thus, a smaller value of $i$ can be used for better efficiency.

### C.3  More Cases

Here we show more cases on test sets to further verify the ability of FABind in finding the correct pocket for unseen protein and docking at the atom level. From Fig. 3(a), in PDB 6N93, the protein is unseen in the training set, and only the predictions of FABind, E3Bind and TankBind are in the right pocket, among which FABind predict the most accurate binding pose (RMSD 2.7Å). From Fig. 3(b), in PDB 6JB4, though every method correctly finds the native pocket, FABind predicts the most accurate ligand conformation (RMSD 1.9Å).

## D  Broader Impacts and Limitations

**Broader Impacts.** Developing and maintaining the computational resources necessary to conduct AI-based molecular docking requires considerable resources, which may lead to a waste of resources.

**Limitations.** In FABind, we represent the protein structure at the residue level, assuming the rigidity of the protein. While many existing molecular docking methods adopt a similar protein modeling strategy, we believe that employing atom-level protein modeling and incorporating protein flexibility into the modeling process could yield improved results. FABind is not optimally designed to accommodate scenarios with multiple binding sites, wherein a ligand has many potential binding

sites on a protein. Additionally, it is not well-suited for situations where a ligand can adopt multiple binding conformations within a specific pocket of a protein. Generative modeling [7] (sample-based method) presents a reasonable approach to address these two scenarios. However, due to the scope and limitations of our current work, we have decided to defer these aspects to future research.

