# OpenReview forum: "FABind: Fast and Accurate Protein-Ligand Binding"
_NeurIPS.cc/2023/Conference — NeurIPS 2023 poster_

### Official Review · Reviewer_Vntv · 2023-06-17

**Soundness:** 3 good
**Presentation:** 2 fair
**Contribution:** 3 good
**Rating:** 6
**Confidence:** 4

**Summary:**

The paper presents a new method for protein ligand docking based on regression of the pocket center and the ligand atomic positions. Although the none of the components of is highly novel, they are well justified and tied together providing strong empirical performance with very fast inference speed. Although there are some adjustments to be made during the rebuttal process (see below) once these are completed I believe the paper will be ready for acceptance.

**Strengths:**

Although most of the main components of the method were present in previous regression based docking methods, the authors tied these together adding a few key ingredients (e.g. center constraint, pairwise distance loss, cross attention) managing to significantly increase the performance over previous methods.

**Weaknesses:**

Main weaknesses:
1. Some parts of the method section lack clarity (see next section) and should be improved.
2. As FABind selects the pocket with a regressive approach (unlike for example TANKBind) it seems that it is critically affected by uncertainty such as symmetries in the receptor, as the residues of the symmetrically opposite pockets would receive the same likelihood and a random independent selection of these would cause the predicted pocket center to be approximately in between the two symmetric pockets. These will apply also in other settings where symmetries are not present but the ligand simply has multiple binding sites. These were artificially removed by the test setting by EquiBind (taking only chains within 10A, not possible in reality without knowing already the binding pocket) but are often present in real blind docking scenarios.
3. Although the method is repeatedly presented as end-to-end, the approach seems to be disjoint with the pocket and docking phases with separate losses that are (almost) independent

Some claims need to be adjusted:
1. line 72: the authors claim improvements in performance based on the mean ligand RMSD metric (the only place in the manuscript where this metric is actually highlighted), a metric which is known not to be reflective of docking performance.
2. line 341: “FABind outperforms existing methods by a large margin with higher efficiency.” “large margin” in performance seems an unjustified claim based on the results (in many cases on par or worse than DiffDock or other methods)

**Questions:**

There are some critical steps of explanation of the method that seems to be missing from both paper and appendix, for example:
1. How is the ligand position initialized and placed during docking? During the pocket prediction since there are message-passing layers between the ligand and the receptor how are they placed relative to one another? During docking how are the ligand nodes initialized during docking? For both: is equivariance to the initial orientation of the receptor guaranteed?
2. Do the layers in the docking and pocket prediction share any parameters? Are the gradients propagated during training between the pocket and docking components? If no to both questions, why is the training of the two components joint (a single loss function) and not separate?

Unclear components in the evaluation:
1. In the appendix 1 it is mentioned that approx. 500/19k complexes are further removed because of the ligand size or number of contacts. Does this filtering also affect the test set?
2. Appendix 2: how is the alignment between the ESMFold and the crystal structures performed?
3. Some of the baselines have reported different numbers compared to their peer-reviewed manuscripts (e.g. TANKBind, DiffDock), is it because of a slightly different test set? If not I recommend the authors to clarify the situation with the authors of the respective publications.
4. A detailed description of the hyperparameter selection process should be added to the supplementary material

**Limitations:**

I was not able to run the provided code to reproduce the results of the paper due to a lack of documentation.

---

> ### Author Rebuttal · Authors · 2023-08-09
>
> Thanks for your time and the review, and the insightful comments. We provide the following responses to your concerns.
>
> > **W1&2: Multiple binding sites**
>
> Thanks for pointing out this important factor. We acknowledge the potential challenges with multiple pockets in the receptor.  In current process, we follow EquiBind to solve the majority of this issue by only keeping the protein chains which have an atom within a 10Å radius of any ligand atom. We show some cases with symmetric proteins in Figure 2 of the uploaded PDF, in which both our method and DiffDock can find the correct pocket. In addition, we can also use external tools to recognize the symmetric proteins (even without knowing the pocket information) in advance and process it by only keeping one of its symmetric parts as input. We want to emphasize that our method is primarily designed to predict the pocket and ligand pose where the binding strength is strongest, which aligns with the data perspective of the PDBBind dataset that emphasizes the strongest natural binding pose to a given site. While symmetries and multiple binding sites may pose challenges, our method reflects the natural objective of identifying the most probable binding configuration. Besides, we would like to highlight: (1) comparison with sample methods, although we may not model the distribution of binding sites like sample methods, validating these sites in practice is also not easy; (2) our method achieves fast speed with also good performance, this is a crucial factor in large-scale virtual screening.
>
> > **W3 & Q1.2: End-to-end training for both pocket prediction and docking tasks**
>
> Sorry for any confusion. We indeed jointly trained one model (all with FABind layer) for both pocket prediction and docking tasks using the comprehensive training loss as described in the paper Section 3.4. The model architecture (FABind layer) shared same structure between the two parts (but not share parameters), and the embeddings are connected. Therefore, the gradients can propagate between these two components, allowing for mutually benefiting from each other. In this way, we streamlined the process and achieved a seamless, end-to-end process for protein-ligand docking prediction.
>
> > **Adjust claims**
> >
>
> Thanks for pointing out these inappropriate claims. We will adjust them in the revised version.
>
> > **Q1.1: Ligand position initialization and equivariance**
> >
>
> We first use Merck Molecular Force Field (MMFF) in RDKit to generate random low-energy ligand conformation, denoted as $C_1$ . Then we apply uniform random rotation on $C_1$ and get $C_2$. Then, for the pocket prediction module, we translate $C_2$ to the center of the protein (i.e., origin of coordinates as we move the protein to the origin of coordinates) to avoid pocket information leakage. For the docking module, we translate $C_2$ to the predicted pocket center by the pocket prediction module.
>
> The equivariance to the initial orientation of the receptor is also guaranteed. For the pocket prediction, as described in paper Section 3.3, we use invariant features (logits) from FABind layer to classify each residue as either belong or not belong to the pocket. Then we calculate the center of these classified residues and take it as the predicted pocket center. So if the initial receptor is translated or rotated, the predicted pocket center will also be translated or rotated accordingly, ensuring the equivariance. For the docking module, the equivariance is naturally guaranteed by the FABind model as it is a variant of EGNN.
>
> > **Q2.1: Data filtering**
> >
>
> This filtering doesn’t affect the test set. It only filters some complexes from the training set.
>
> > **Q2.2: Alignment between the ESMFold and the crystal structures**
> >
>
> Thanks for this question. We follow the DiffDock to align the ESMFold predicted structure to the crystal structures to obtain the “ground truth” docked structures for apo-structure docking experiments. More details can be found in DiffDock Appendix Section D.2. Briefly speaking, the alignment process involves using the Kabsch algorithm with exponential weights to align receptors' residues. For each receptor, a weight is assigned based on the minimum distance to a ligand atom, with closer residues receiving higher weight, and the smoothing factor $\lambda$ is individually selected within the range [0, 1] to best preserve distances. The L-BFGS-B method is used to minimize the $\lambda$.
>
> > **Q2.3: Reported baseline performance**
> >
>
> Thanks for pointing out this. There is no difference in the test sets between the compared baselines and our work. The results of Table 1 and 2 mainly come from the E3bind, including traditional docking methods: QVina-W, GNINA, SMINA, GLIDE, Vina, as well as DL-based methods: EquiBind, TankBind, E3Bind. For the DiffDock results, as the results of sample-based method are unstable (both issues from DiffDock github and our experiments have found this), we reproduce the results using its pre-trained checkpoint. Specifically, we run the DiffDock inference codes with three random seeds, and report the average results.
>
> > **Q2.4: Hyperparameter selection**
> >
>
> Thanks for the question. To clarify, some hyperparameters were set following previous works to ensure consistency and comparability. For the remaining hyperparameters, we employed a grid search method to explore the optimal values that would yield the best performance. We will include a comprehensive description of the hyperparameter selection process in the supplementary material.
>
> > **L1: Reproductivity**
> >
>
> Sorry for the lack of documentation that hindered to run the provided code. We have made a documentation to run the code, and updated the code to AC according to the rebuttal rule. Please contact AC if needed.

---

> > ### Comment · Reviewer_Vntv · 2023-08-15
> > **Response to rebuttal**
> >
> > Thank you very much for the response that addresses many of my concerns. Regarding the first point on pocket finding: while getting rid of other chains is a sensible solution when one knows (approximately) the correct binding site, this does not apply to the blind docking/pocket finding scenario (where such chains are not known in advance). Moreover, often ligands bind with contacts to multiple chains rendering this procedure obsolete. Therefore to have a reliable pocket prediction method I suggest the authors to derive ways of overcoming this averaging problem. However, overall I believe that the paper deserves acceptance so I have raised my score to 6

---

> > > ### Author Response · Authors · 2023-08-16
> > >
> > > Thanks for your constructive feedback and the acknowledgement. We concur with your observation regarding the challenges of blind docking/pocket finding scenarios and the potential drawbacks of removing other remote chains.
> > > In light of your suggestions, we agree that improvements and more advanced methods should be considered for future version. A potential approach maybe to leverage the logits from the classification head to perform clustering. By identifying clusters with higher logits (which could be one or more), we could potentially achieve a more reliable pocket prediction so to better handle scenarios where ligands bind to multiple binding sites. We believe this scenario you mentioned should be highly focused in the future work.

---

### Official Review · Reviewer_5BP6 · 2023-06-28

**Soundness:** 3 good
**Presentation:** 3 good
**Contribution:** 3 good
**Rating:** 5
**Confidence:** 3

**Summary:**

This paper proposes an end-to-end model, FABind, which is a pocket prediction and docking end-to-end model considering the prediction accuracy and speed at the same time. By gradually adding the predicted pocket into the docking process, the model improves protein-ligand binding while minimizing differences between training and inference. An equivariant layer is also added in the model. According to empirical tests, FABind performs significantly better than the alternatives.

**Strengths:**

1. This work considers limitations for sampling-based methods and regression-based methods and combines pocket prediction and docking to achieve accurate and fast protein-ligand binding.
2. This approach employs a particular ligand to identify the precise pocket that the ligand binds to, as compared with conventional pocket prediction, which just uses protein as an input and predicts numerous probable pockets.

**Weaknesses:**

1. Why you choose 10 \angstrom as the cut-off distance, have you tried other distances as the hyperparameter?
2. The comprehensive training loss includes the pocket losses and docking losses, which each have two different kinds of losses, there are many weight hyperparameters that should be determined, how you choose the weight hyperparameters? The pocket prediction and docking seem to be two different tasks? Did you train only one model for these two tasks by the comprehensive training loss?
3. In Table 1., the performance of the proposed model, FABind seems not good enough, as the DiffDock(40) has many superior results, what is the reason? But for the flexible blind self-docking, FABind performs much better than DiffDock, what cause these differences?
4. You have pointed out the limitations for the sampling-based methods and regression-based methods, low efficiency and decreased accuracy, and the variation in protein sizes, I guess it may be better to show what is based on sampling-based methods, and what is based on the regression-based methods in the results, Table 1 and 2. Some models are tested on CPU, the comparison seems not fair, as you add the operation of refinement, which will increase the time. How you tackle the problem of the variation in protein sizes as you have mentioned?

**Questions:**

See above.

**Limitations:**

Yes.

---

> ### Author Rebuttal · Authors · 2023-08-09
>
> Thanks for your effort and the detailed questions about our work. We try our best to provide the following response to help address your concerns.
>
> > **W1: Cut-off distance**
>
> We follow TankBind [1] and E3Bind [2] to set the cutoff to 10Å, which balances computational efficiency and the ability to capture relevant interactions.
>
> According to your suggestion, we conducted an ablation study to explore the impact of different cut-off distances. We experimented with 6Å, 10Å, and 14Å, and the results are shown in the Table 2 of the uploaded PDF. Our findings revealed that the 6Å cut-off resulted in the worst performance, while 10Å yields slight better results than 14Å. We opted for 10Å as the cut-off distance to enhance efficiency.
>
> > **W2: Loss weight hyperparameter and joint training**
>
> Thanks for the detailed questions, the answers are as follows.
>
> 1. **Selection of Weight Hyperparameters**: For the loss weight, we conducted a grid search around hyperparameters that allowed losses to be on a similar scale. Specifically, we fixed the weight for the pocket classification loss at 1 and searched for the pocket center constraint loss weight $\alpha$ in the range of [0.1, 0.2, 0.3, 0.4, 0.5]. For the remaining hyperparameters $\beta$ and $\gamma$ in the docking loss, as well as the overall ratio between pocket loss and docking loss, we set the search range as [0.25, 0.5, 1.0, 2.0, 4]. This approach allowed us to find an optimal balance among the different components of the loss function. We choose the model with best valid performance for the final test.
> 2. **Training Approach for Pocket Prediction and Docking**: In previous works, pocket prediction and docking tasks are separately trained. For instance, TankBind and E3Bind utilize external tools P2Rank for offline pocket prediction.  However, in our approach, we jointly trained one model (all with FABind layer) for both tasks using the comprehensive training loss as described in the paper Section 3.4. The model architecture (FABind layer) shared the same structure between the two parts, and the embeddings are connected, allowing the two components to mutually benefit from each other through joint optimization. In this way, we streamlined the process and achieved a seamless, end-to-end process for protein-ligand docking prediction.
>
> > **W3: Comparison with DiffDock**
>
> For the DiffDock(40) [3] performance in terms of the <2Å metric, it's essential that DiffDock's confidence model is trained specifically for the classification task of whether the ligand RMSD is below 2Å or not. This specialized training is a reason why DiffDock performs better on this specific metric. Moreover, our FABind model exceeds the performance of DiffDock on 10 sampled poses with much higher efficiency.
>
> For the second part, we think you mean that in the flexible blind self-docking on unseen receptors experiment, FABind performs much better than DiffDock. We attribute this result to the following aspects:
>
> 1. We have carefully modeled pocket prediction module, achieving excellent pocket prediction through jointly optimizing the pocket prediction and docking problems. Compared to DiffDock, our model is more accurate in pocket prediction (in Table 2 of the paper, Centroid Distance <2Å metric, FABind 57.6 % v.s. DiffDock(40) 43.4%) with better generalization, possibly contributing to the superior performance.
> 2. The confidence model of DiffDock primarily focuses on optimizing ligand poses towards achieving <2Å accuracy. This specific focus allows DiffDock to excel in the <2Å metric, but it may not generalize well to unseen receptors.
>
> We will add more discussion about this in the later version.
>
> > **W4: Classification of methods, time comparison, variation in protein sizes**
>
> Thank you for raising these insightful questions, which we address as follows:
>
> 1. **Classification of Methods**: To clarify the type of the methods used, we will include an additional column in Tables 1 and 2 to indicate whether each model is sample-based or regression-based. Briefly speaking, among the DL based methods, only DiffDock is sample-based methods while the others are all regression-based methods.
> 2. **Computational Time Comparison**: We strive for a fair comparison by utilizing GPU test where available and resorting to CPU only when GPU implementation is absent. Most of the traditional methods listed in the top half of the table do not have GPU implementations, so CPU testing is the only choice. Further, we have demonstrated the difference in computational speed due to iterative refinement in Appendix Table 3. This clearly shows that the refinement operation does not compromise FABind's status as a fast approach. Besides, we follow previous works such as TankBind and E3Bind that the comparison is the same as what they adopted.
> 3. **Handling Variation in Protein Sizes**: PDBBind dataset consists of a wide variation in protein sizes, ranging from 28 to 3012 amino acids. This variation, if fed directly to the docking module, could pose challenges in capturing key contact surface information and would consume significant GPU memory. Our solution involves introducing a pocket prediction module. Regardless of the whole protein's size, the docking module only models the pocket part, consisting of a small number of amino acids, such as a little over a hundred. Models that perform docking directly on the whole protein, such as Equibind and DiffDock, can lead to substantial computational costs for larger proteins.
>
> **References**
>
> [1] Lu, Wei, et al. "Tankbind: Trigonometry-aware neural networks for drug-protein binding structure prediction." *Advances in neural information processing systems* 35 (2022): 7236-7249.
>
> [2] Zhang, Yangtian, et al. "E3bind: An end-to-end equivariant network for protein-ligand docking." *arXiv preprint arXiv:2210.06069* (2022).
>
> [3] Corso, Gabriele, et al. "Diffdock: Diffusion steps, twists, and turns for molecular docking." *arXiv preprint arXiv:2210.01776* (2022)

---

> > ### Comment · Reviewer_5BP6 · 2023-08-17
> >
> > Thank you for your explanation of the difference in results between Table 1 and Table 2. There are two cases provided in the manuscript, the large unseen protein PDB 6NPI and the relatively small protein PDB 6MO2. Can the results of these two examples be generalized to others large and small proteins? If so, (1) why on the large unseen protein,  FABind succeeded while others failed, and on the small protein, FABind can make better predictions? (2) Are there any lab experiments to validate some conclusions or observations?

---

> > > ### Author Response · Authors · 2023-08-17
> > >
> > > Thank you for your careful assessments of the results and the cases. To look at generalizability of our FABind on proteins of different sizes, we conducted an experiment on unseen proteins with different sizes, and we evaluated on FABind, DiffDock and TankBind for comparison. We categorized proteins based on their 1D FASTA sequence length, using the mean length $452$ of the unseen test set as a threshold to distinguish the small and large proteins.
> > >
> > > The results are presented in the following Table 1 and 2 (we use “CentDist” in tables to represent Centroid Distance, which measures the ability of the model to find the correct binding site). We can find that our **FABind consistently performs better on both small and large proteins**, indicating its ability to generalize to other proteins irrespective of their sizes. We attribute the reason for the **superior performance of FABind to our more accurate pocket prediction module** (can be demonstrated by the CentDist values), which is jointly optimized with docking module. Compared to DiffDock with 40 sample times, FABind’s pocket prediction accuracy is slightly better on small proteins and greatly better on large proteins. This enhanced accuracy in pocket prediction subsequently leads to a more precise RMSD.
> > >
> > > | Method/Metric | RMSD 25%↓ | RMSD 50%↓ | RMSD 75%↓ | RMSD Mean↓ | RMSD < 2Å↑ | RMSD < 5Å↑ | CentDist 25%↓ | CentDist 50%↓ | CentDist 75%↓ | CentDist Mean↓ | CentDist < 2Å↑ | CentDist < 5Å↑ |
> > > | --- | --- | --- | --- | --- | --- | --- | --- | --- | --- | --- | --- | --- |
> > > | TankBind | 2.7 | 4.4 | 7.0 | 7.7 | 7.5 | 63.4 | 1.1 | 2.1 | 3.8 | 5.8 | 49.5 | 79.6 |
> > > | DiffDock (40) | 2.2 | 6.4 | 13.8 | 9.5 | 23.7 | 39.8 | 0.7 | 2.6 | 9.8 | 7.4 | 46.2 | 61.3 |
> > > | FABind | 2.0 | 3.1 | 9.4 | 7.5 | 26.9 | 60.8 | 0.7 | 1.5 | 4.6 | 5.6 | 58.1 | 76.3 |
> > >
> > > Table 1: Performance of different methods on unseen small proteins.
> > >
> > > | Method/Metric | RMSD 25%↓ | RMSD 50%↓ | RMSD 75%↓ | RMSD Mean↓ | RMSD < 2Å↑ | RMSD < 5Å↑ | CentDist 25%↓ | CentDist 50%↓ | CentDist 75%↓ | CentDist Mean↓ | CentDist < 2Å↑ | CentDist < 5Å↑ |
> > > | --- | --- | --- | --- | --- | --- | --- | --- | --- | --- | --- | --- | --- |
> > > | TankBind | 3.7 | 6.6 | 9.9 | 12.4 | 4.1 | 40.8 | 1.5 | 2.9 | 6.5 | 10.1 | 36.7 | 67.3 |
> > > | DiffDock (40) | 3.7 | 8.7 | 25.6 | 17.7 | 2.0 | 38.8 | 1.6 | 3.3 | 25.5 | 15.0 | 36.7 | 57.1 |
> > > | FABind | 2.7 | 3.6 | 7.5 | 8.5 | 12.2 | 57.1 | 1.2 | 1.5 | 4.7 | 7.0 | 55.1 | 75.5 |
> > >
> > > Table 2: Performance of different methods on unseen large proteins.
> > >
> > > To further validate our conclusions, we sought to provide additional evidence through computational means, as conducting wet lab experiments was beyond the scope of our current study (we are not able to provide such experiments due to resource unavailable). We employed the scoring function of AutoDock Vina to calculate the docking score. The results in Table 3 reveal that FABind achieved lower docking scores compared to baseline methods, indicating better performance.
> > >
> > > | Method/PDB id | 6o5u | 6j9w | 6k3l |
> > > | --- | --- | --- | --- |
> > > | TankBind | -4.36754 | 24.37799 | -5.00976 |
> > > | DiffDock (40) | -2.84643 | -1.32461 | -7.72575 |
> > > | FABind | -5.02038 | -9.63302 | -8.24568 |
> > >
> > > Table 3: Vina score of different methods on selected cases.
> > >
> > > For clarity, the Vina scoring function provides an estimate of the binding affinity between the ligand and the protein. A lower Vina score indicates a more favorable binding interaction, suggesting that the ligand and protein fit together well.
> > >
> > > We hope these explanations can answer your questions and we are grateful for your further feedback.

---

> > > > ### Comment · Reviewer_5BP6 · 2023-08-17
> > > >
> > > > Thanks for providing this experiments and explanations. I have no problems at this point. Considering the speed and accuracy of this model, I increase my score.

---

> > > > > ### Author Response · Authors · 2023-08-18
> > > > >
> > > > > Thanks for your thoughtful feedback and acknowledgment.

---

### Official Review · Reviewer_N7PK · 2023-07-15

**Soundness:** 3 good
**Presentation:** 3 good
**Contribution:** 3 good
**Rating:** 6
**Confidence:** 4

**Summary:**

This paper proposed a graph model named FABind for protein-ligand binding prediction. The major difference from existing work is that FABind integrate both protein pocket prediction and protein-ligand binding in a single model. The pocket prediction subtask is modeled as a binary classification problem, and the docking is a regression problem regulated with distance map based loss. The model was trained and tested using PDBbind v2020 data. Comparison with force-field based tools and GNN-based methods show the advantages of FABind. Furthermore, FABind's inference runs much faster than most of the benchmarks. Case studies are also provided in the main text and supplementary file.

**Strengths:**

The major contribution of this work is the combination of protein pocket prediction and protein-ligand binding in one single GNN model.  paper is well written as well. I found it is very easy to follow.

**Weaknesses:**

The protein-ligand complexes in training and test should be better explained. I am wondering whether negative protein-ligand pairs are used in both processes. Particularly, I am interested in seeing the performance for negative protein-ligand pairs. For protein pocket prediction, what would the model infer if a protein is undruggable, i.e.,  there is no native pocket on its smooth surface. In a different scenario, a protein could have multiple binding pocket; how would FABind behave in such cases?

**Questions:**

Please see limitation above.

**Limitations:**

The authors should discuss more about the use of negative protein-ligand pairs in model fitting and evaluation. Also, different scenarios of protein pockets should be considered in evaluations.

---

> ### Author Rebuttal · Authors · 2023-08-09
>
> Thank you for your insightful feedback. We take the opportunity to clarify the aspects you have raised.
>
> > **The protein-ligand complexes in training and test should be better explained.**
> >
>
> The dataset details are in the appendix, we utilized the PDBBind dataset for our study. Each data sample provides the naturally docked complex structure. The training, validation, and test sets were split based on time, with the test set containing only complexes discovered in 2019 or later, while the train and validation sets use complexes before 2019. More details on this can be found in Appendix Section 1.1.
>
> > **I am wondering whether negative protein-ligand pairs are used in both processes. Particularly, I am interested in seeing the performance for negative protein-ligand pairs.**
> >
>
> Our approach models docking as a regression task, and evaluation metrics are the distance between the predicted and ground ligand atoms/center, such as RMSD and Centroid Distance. Unlike classification tasks, constructing negative examples through a random combination of ligand and protein is not feasible in this context, as determining the ground truth for randomly combined pairs is ambiguous, and different data samples are not in the same coordinate system.
>
> > **For protein pocket prediction, what would the model infer if a protein is undruggable, i.e., there is no native pocket on its smooth surface.**
> >
>
> Thanks for this question, which is a really interesting point. We explored this by selecting an undruggable target, Myc [1], known for features that render it undruggable, such as the lack of medicinally binding pockets and its relatively smooth surface. We performed inference on this example by randomly selecting some ligands from PDBBind, and the results are included in Figure 1 of the uploaded PDF. We can see that FABind docks these ligands to a position but it is not a pocket. Therefore, the visualization would clearly show for undruggable protein, there would be no correct docking poses by our model.
>
> > **In a different scenario, a protein could have multiple binding pocket; how would FABind behave in such cases?**
> >
>
> In nature, given a ligand, it may binds to different pockets of a protein. In this work, we mainly focus on the most probability binding pocket that a ligand would bind (the strongest binding strength) to, where the PDBBind dataset is also processed in this way. This is also the case we care mostly in real scenarios.
>
> The scenario you mentioned mostly lies in the proteins that are symmetric or multimeric. In these cases, the ligand can equally bind to any of these symmetric pockets as correct one. We solve the majority of this issue by only keeping the protein chains which have an atom within a 10Å radius of any ligand atom. Such cases and the comparison with DiffDock [2] can be seen in Figure 2 of the uploaded PDF. Both our method and DiffDock can find the correct pocket. Besides, we can use external tools to recognize the symmetric proteins and post-process the results.
>
> **References**
>
> [1] He, Huiqin, et al. "Big data and artificial intelligence discover novel drugs targeting proteins without 3D structure and overcome the undruggable targets." *Stroke and Vascular Neurology* 5.4 (2020).
>
> [2] Corso, Gabriele, et al. "Diffdock: Diffusion steps, twists, and turns for molecular docking." *arXiv preprint arXiv:2210.01776* (2022)

---

### Official Review · Reviewer_oHVp · 2023-07-15

**Soundness:** 2 fair
**Presentation:** 3 good
**Contribution:** 2 fair
**Rating:** 5
**Confidence:** 3

**Summary:**

In this paper, a protein-ligand binding algorithm based on the E3NN framework, named FABind, is proposed for docking molecules and proteins. The algorithm firstly predicts the pockets on the interface of the proteins, and then utilizes the predicted pockets to dock with the molecule ligand, and the algorithm results show that FABind demonstrates superior performance to TANKBIND, E3BIND, and other methods on protein-molecule docking datasets.

**Strengths:**


1. predict pocket in docking framework is reasonable.

2. The experimental results show that FABind outperforms TANKBIND, E3BIND and realizes a new sota.


**Weaknesses:**

1. FABind used EGNN to update atom coordinates, although EGNN is equivariant neural network. however EGNN doesn't maintain rigid constraint for protein. It's the major weakness when using EGNN on rigid docking problem.

2. it's not clear why the neural network need a separate interfacial message passing module because you have already use cross attention neural network. you can use attention as a soft interface.


**Questions:**


1. In the right of Figure 1, ligand is n*3, receptor is n*3, but pair representation is n*m*3, the dimension is a little bit confusion.

2. can FABind solve atom clash？

3. what the performance on pocket prediction?


**Limitations:**


1. No release code

2. no wet-lab experiments

---

> ### Author Rebuttal · Authors · 2023-08-09
>
> Thanks for your thorough review and invaluable suggestions. We provide the following responses to address your concerns.
>
> > **W1: FABind used EGNN to update atom coordinates, although EGNN is equivariant neural network. however EGNN doesn't maintain rigid constraint for protein. It's the major weakness when using EGNN on rigid docking problem.**
> >
>
> Thank you for pointing out the potential limitation of using EGNN for rigid docking problems. We recognize that this might be a theoretical challenge. However, in our implementation, we employed a masking strategy where we only update the ligand's coordinates and keep the protein's coordinates fixed during each pass through the model. This ensures the rigidity of the protein structure. We will add detailed description of this in the later version.
>
> > **W2: It's not clear why the neural network need a separate interfacial message passing module because you have already use cross attention neural network. you can use attention as a soft interface.**
> >
>
> Thank you for highlighting the potential role of cross attention in extracting a soft interface. In our work, we choose to incorporate an interfacial message passing module for the following reasons:
>
> 1. **Focused Updating**: By constructing a hard interface, we can update independent and interfacial structures separately. Utilizing cross attention alone for soft interface updating might lead to the model altering structures far from the contact surface. This is not desirable in our case, as the docking process mainly occurs at the contact surface. Our approach allows for more targeted updates that concentrate on the key areas involved in docking.
> 2. **Dynamic Structure Maintenance**: While we dynamically update the structure, the hard interface is also updated accordingly. By stacking the interfacial message passing layer multiple times, we force the docking module to dynamically capture the interaction.
>
> > **Q1: In the right of Figure 1, ligand is n*3, receptor is n*3, but pair representation is n*m*3, the dimension is a little bit confusion.**
> >
>
> Sorry for the inconsistency in Figure 1, where the receptor should be represented as $m^3$, not $n^3$. Thanks for your careful review and we will correct this typo in the revised version.
>
> > **Q2: Can FABind solve atom clash？**
> >
>
> Thank you for your question regarding the ability of FABind to solve atom clash.
>
> 1. Within the ligand, atom clashes can be resolved through post-optimization. Specifically, post-optimization ensures that there are no clashes between atoms by constraining the distance between atoms by bond lengths. The performance of post-optimization is detailed in Appendix Table 2.
> 2. We acknowledge the importance of addressing this issue. Though our loss function does impose a soft constraint on the distance between ligand atoms and protein alpha carbons, it cannot guarantee the complete absence of clashes between the ligand and the protein. This is also a limitation of most current models. We will explore methods to better handle this in the future work.
>
> > **Q3: What the performance on pocket prediction?**
> >
>
> We have conducted the pocket prediction experiments and analyses in the paper Section 5.1. We compare our pocket prediction module with DL-based docking methods including TankBind [1] and E3Bind [2]. We also analyze the effect of combining ligand conformation and center constraint (regression loss) to the pocket prediction performance.
>
> **References**
>
> [1] Lu, Wei, et al. "Tankbind: Trigonometry-aware neural networks for drug-protein binding structure prediction." *Advances in neural information processing systems* 35 (2022): 7236-7249.
>
> [2] Zhang, Yangtian, et al. "E3bind: An end-to-end equivariant network for protein-ligand docking." *arXiv preprint arXiv:2210.06069* (2022).

---

> > ### Comment · Reviewer_oHVp · 2023-08-17
> > **Response**
> >
> > Thanks for your response. The pocket is extracted from native docking structure or not? If there are multiple pockets, does your method can solve this scenario. It's interesting to see evaluation on multiple-conformation scenarios.

---

> > > ### Author Response · Authors · 2023-08-18
> > >
> > > Thanks for your further response and insightful questions.
> > >
> > > > **The pocket is extracted from native docking structure or not?**
> > > >
> > >
> > > During training, we use the pocket from the native docking structure as the ground-truth label for learning (since in the training data, the protein structure is the docked structure from Protein Data Bank). During inference, the pocket is directly predicted by our model (described in Section 3.3, lines 200-207) and then used for docking. We have also included an experiment in Appendix Section 2, the Apo-structure docking, where the protein structure is predicted by ESMFold [1], which may differ from the native docking structure. We show that our method still achieves good performance compared to other methods.
> > >
> > > > **If there are multiple pockets, does your method can solve this scenario? It's interesting to see evaluation on multiple-conformation scenarios.**
> > > >
> > >
> > > Thanks for this question. For a given protein, there may exist multiple binding sites. Our pocket prediction module takes both protein and ligand information for pocket prediction, which then serves as input to the docking module. Therefore, if different ligands bind to different pockets of the same protein, it can be naturally solved as the input ligand information/features are different for such cases (for each ligand, our method will predict its best-fitted pocket and form a unique ligand-pocket pair for docking). For the scenario of the same ligand binding to different pockets of a specific protein, the most important case is that such proteins are symmetric or multimeric proteins. In such proteins, the ligand can equally bind to any of these symmetric pockets as the correct one. We can simply take any of the symmetric sections of the protein to predict the correct pocket. Currently, we address the majority of this issue by only keeping the protein chains that have an atom within a 10Å radius of any ligand atom. Such cases and the comparison with DiffDock [2] can be found in Figure 2 of the uploaded PDF, in which both our method and DiffDock can find the correct pocket. However, we believe such a scenario should be paid in large efforts in future research.
> > >
> > > By the way, we apologize for overlooking your concern regarding code availability in the previous “Limitations” section. We actually have included the code in the supplementary material and a more detailed version has been provided to the AC according to the rebuttal guidelines. Please reach out to the AC if needed.
> > >
> > > **References**
> > >
> > > [1] Lin, Zeming, et al. "Evolutionary-scale prediction of atomic-level protein structure with a language model." *Science* 379.6637 (2023): 1123-1130.
> > >
> > > [2] Corso, Gabriele, et al. "Diffdock: Diffusion steps, twists, and turns for molecular docking." *arXiv preprint arXiv:2210.01776* (2022)

---

### Official Review · Reviewer_wgwB · 2023-07-18

**Soundness:** 2 fair
**Presentation:** 2 fair
**Contribution:** 3 good
**Rating:** 6
**Confidence:** 3

**Summary:**

**Overview**. The paper proposes a model for flexible blind docking, where, without knowledge of the protein pocket, the model must predict the configuration (translation, rotation, conformation) of a given ligand bound to a given protein. The authors categorise prior deep-learning methods as either regression- or sampling-based, with, e.g., TankBind, EquiBind, and E3Bind being regression-based and, e.g., DiffDock being sampling-based. The authors describe a trade-off: while sampling-based methods are accurate, they are slow in inference time; and while regression-based methods are fast in inference time, they are inaccurate. The authors propose a fast regression-based method that they claim rivals the accuracy of sampling-based methods. The method breaks the task of blind docking into two subtasks, **pocket prediction** and **docking**, and uses a learned equivariant **representation** of proteins and ligands to perform them. The model is reported to have comparable performance to DiffDock (SOTA) in blind docking, and overall better performance than DiffDock when unseen proteins are used.

**Representation**. Their model represents proteins as amino acid residue graphs with $C_\alpha$ coordinates, edges between $8 Å$-away residues, and node features generated by ESM-2. Ligands are represented as atom graphs with edges for every bond a node features generated by TorchDrug. Pair embeddings between protein nodes and ligand nodes are computed with an outer product of the protein features and ligand features (following E3Bind). The authors propose a layer, called FABind, to update the coordinates and features of the protein and ligand graphs, which consists of three components. The first component, Independent message passing, updates the protein and ligand graphs' coordinates and features by performing Equivariant Graph Convolution (from EGNN) within the graphs separately. The second component, Cross Attention Update, updates the protein and ligand graphs' features (as well as the pair embeddings) by applying attention with message passing over all residue-atom pairs. The third and final component, Interfacial Message Passing, updates the coordinates and features of nodes in the protein and ligand graphs that are on a contact surface, using attention and message passing on the contact surface.

**Pocket prediction and Docking**. Using the updated protein and ligand graphs, binary classification is done on the protein graph to determine which residues belong to the binding pocket using a binary cross-entropy loss. Then, using Gumbel-Softmax, a weighted mean of the decided pocket residues is computed as the predicted centroid of the pocket, and finally, the Huber loss between the predicted centroid and the true centroid is computed as a loss. The ligand coordinates and features are fed through the FABind layer several times ("iterative refinement"). One loss is computed by the Huber distance between predicted and true coordinates. Another loss is computed by the sum of three $L_2$ sub-losses. The first sub-loss is between the true distance matrix (a matrix of distances between pairs of atoms in the ligand) and the distance matrix computed from FABind's predicted ligand coordinates. The second sub-loss is between the true distance matrix and a distance matrix computed with an MLP from FABind's predicted pair embeddings. The third sub-loss is between these two predicted distance matrices. Training is done in a scheduled manner (following so-called teacher-forcing), where the model is first trained for docking with the known pocket; and then, once the pocket prediction is reasonably good, it is used for docking.

**Results**. The model is evaluated on PDBBind 2020, with similar preprocessing to EquiBind, TankBind, and DiffDock. The model is reported to outperform all other models in the percent of predictions with ligand RMSD below 5Å, and all models except DiffDock with ligand RMSD below 2Å. The model is also reported to outperform DiffDock when presented apo structures of the proteins.

**Strengths:**

1. The model is significantly faster than DiffDock (SOTA), while still achieving comparable performance in blind docking.
2. The model is reported to perform better than DiffDock in blind docking with unseen proteins.
3. The model does not require additional geometric constraints as in DiffDock (i.e. group transformation manifolds), making the formulation a bit easier to understand at times.

**Weaknesses:**

1. The results are not significantly better than prior methods: the percent of predictions with a ligand RMSD below 5Å is higher than that of other models. However, the percent of predictions with a ligand RMSD below 2Å is not greater than that of DiffDock. Any ligand prediction with RMSD above 2Å is not very useful in itself (but may be useful for "warm-starting" and speeding up search-based methods).
2. The model seems to assume that there is one unique binding pocket that a given ligand will bind to. In reality, this is not true, as, in addition to an orthosteric binding site (the natural one), there might exist other allosteric binding sites (not the natural one). Therefore, sampling-based methods seem to have the upper hand in this case, as they are able to capture multiple different binding modes.
3. It seems that the model also assumes that there is one unique bound ligand configuration (since this is a regression-based model) which might not be true either.
4. The structure of the paper was a bit difficult follow and the related work section seemed a bit short. I would appreciate a more detailed comparison of this model and other prior models. A detailed explanation of how this model is different from previous models should be included.
5. I could not get the code to run because I could not find the module `metrics_to_tsb`. Please advise.

**Questions:**

1. How does the model perform when independent message passing is removed?
2. How are the external contact surface edges defined? I did not see an explanation.
3. How does the model perform when used together with a searched-based method (i.e. "traditional docking software")?
4. In the apo structure docking experiments, where is most of the flexibility in the receptors (i.e. mostly in the backbone or mostly in the side chains)? Could you show how the model performs when presented with receptors that have most of their flexibility in the backbone (and when most of the flexibility is in side chains)?
5. How does the model handle the existence of multiple possible binding sites and/or multiple possible bound ligand configurations? Could you please show the models in these cases (and compare with DiffDock)?
6. Could you please elaborate on which pre-trained features were used in the protein and ligand graphs?
7. Why were edge features not included (e.g. single and double bonds in ligands)?

**Limitations:**

1. The model treats proteins as rigid, which is not true for a large portion of important proteins (e.g. GPCRs), though a reasonable assumption for many.
2. The model assumes a unique binding site and ligand configuration (there could be multiple possible sites and configurations).

---

> ### Author Rebuttal · Authors · 2023-08-09
>
> Thanks for your insightful comments. We appreciate your time and effort and would like to address the concerns.
>
> > **W1: Comparison with DiffDock**
>
> Thanks for careful comparison. DiffDock is a sample-based method, and each sampling will generate a sampled pose and it employs an extra confidence model to select the most appropriate conformation. The large sampled candidates contributed to its performance.
> We acknowledge the case on <2Å with 40 sampled poses, but would like to point out that FABind achieves comparable/better accuracy at a sample size of 10 and is several orders of magnitude faster(170 times faster). When compared to DiffDock with 40 sampled poses, FABind is slightly worse but over 690 times faster. This efficiency is particularly vital in large-scale virtual screening. Another point is that the confidence model in DiffDock is specifically trained in terms of <2Å metric, this contributes to its good performance when using 40 sampled poses.
>
> In addition, FABind shows better performance than DiffDock(40) in blind docking with unseen receptors(Tab.2) and apo-structure docking(Appendix Tab.1).
>
> Moreover, FABind is agnostic to training algorithm, enabling easy integration with diffusion. From the results in Tab.1, we believe even better results could be achieved when adapted to a sample-based method.
>
> > **W2&W3&Q5: Multiple binding sites/ligand configurations, cases compared with DiffDock**
>
> Thanks for pointing this concern. We agree there maybe multiple binding sites/pockets. However, our method is mainly designed to predict pocket and ligand pose with the maximal probability(strongest binding strength). This also aligns with data perspective, as PDBBind dataset focuses on the strongest natural binding pose to the site. Besides, we give some discussions about these special cases with multiple binding sites and multiple conformations.
>
> (1) **Multiple binding sites.** The most important case is that such proteins are symmetric or multimeric proteins. For these cases, the ligand can equally bind to any of these symmetric pockets as correct one. We can simply take any of symmetric part of the protein to search correct pocket. In current process, we solve majority of this issue by only keeping protein chains which have an atom within a 10Å radius of any ligand atom. Such cases and the comparison with DiffDock can be found in Figure 2 of the uploaded PDF, in which both our method and DiffDock can find the correct pocket. In addition, we can also easily use external tools to recognize symmetric proteins and post-process the results in case the regression method has this limitation.
>
> (2) **Multiple bound conformations.** Regression methods truly suffer from the mean prediction if there are multiple bound conformations. It’s true that sample methods may generate multiple poses, but the validation is also not easy. And we usually care about the one with lowest energy, which aligns with current training data and is what we focus on.
>
> > **W4: Paper structure and related works**
>
> We apologize for any confusion. The logic in the method is to first introduce FABind layer, which is used in the whole framework that contains pocket prediction and docking prediction we introduced next. Finally we summarize with a pipeline(training method) subsection. For related works, due to the space limitation, we’ll give more comparison in later revision.
>
> > **W5: `metrics_to_tsb`**
>
> Sorry for oversight. This module only stores metrics for TensorBoard and can be removed. We have updated the code to AC according to the rebuttal rule. You can contact AC if needed.
>
> > **Q1: Remove Independent Message Passing(IMP)**
>
> IMP is crucial as it represents internal force field updates within ligand or protein. Removing it would lead to an inability to update node coordinates that are distant from the contact surface. We have conducted an ablation in the Tab.3 of the uploaded PDF to prove this.
>
> > **Q2: External contact surface edges**
>
> Due to space constraints, we have placed the definition in Appendix Sec 1.4. For the edges at ligand-protein interface, we employed a cut-off distance less than 10Å to determine whether an edge should be constructed between two points.
>
> > **Q3: Together with traditional docking software**
>
> Thanks for the advice. We conduct the following study with SMINA and our FABind. We use FABind to predict the pocket first, and then feed into SMINA for docking. Results are in Tab.1 of the uploaded PDF. When without specifying our FABind predicted pocket(docking on entire protein), SMINA performs less accurate. In contrast, with our predicted pocket, the accuracy is improved, but still worse than FABind. This proves that our FABind can predict effective pockets and the docking module is superior than SMINA.
>
> > **Q4: Apo-structure docking**
>
> There might be some misunderstandings about the flexibility in this setting. The apo-structure docking, proposed by DiffDock, assumes protein is still rigid, meaning there is no flexibility in backbone and side chains. Compared to the holo-structure(bound), which is provided by PDBBind, apo-structure(unbound) is computationally generated(by ESMFold) and aligned with original holo-structure. We follow the same setting as DiffDock and will clarify this point in the revised version.
>
> > **Q6: Pre-trained feature usage**
>
> For protein graph, we use pre-trained ESM-2 features to initialize. For ligand graph, no pre-trained features are used. The node feature is a 56-dimensional vector related to atom, degree, etc. More details about features in protein and ligand can be found in Appendix Sec 1.3.
>
> > **Q7: Edge features not included**
>
> In graph learning, the use of edge features is not universally adopted. Some methods incorporate edge features, while others not. There is no consensus in scientific community about which approach is superior. In our study, we follow TankBind and E3Bind that do not use edge features. We are open to explore edge features in future work.

---

> > ### Comment · Reviewer_wgwB · 2023-08-18
> >
> > I will update my score to indicate accept, mainly based on the merit of the model's significant speed and the authors' rebuttal. However, I have a few comments below.
> >
> > > **W2&W3&Q5: Multiple binding sites/ligand configurations, cases compared with DiffDock**
> >
> > I do see the authors' point: since PDBBind "focuses on the strongest natural binding pose to the site", it is reasonable to seek to "predict pocket and ligand pose with the maximal probability (strongest binding strength)". However, it is true, e.g., that finding allosteric (rather than the normal orthosteric) binding sites is of notable interest in drug discovery.
> >
> > > **Q3: Together with traditional docking software**
> >
> > What is the explanation for FABind performing better than FABind + SMINA?
> >
> > > **Q4: Apo-structure docking**
> >
> > I understand that both FABind and DiffDock consider the protein to be rigid. My question is: how does the difference in structure between the holo form (from PDBBind) and apo form (from ESM-fold) affect FABind's prediction? If the difference is mostly in the side-chains (i.e. not in the backbone), then I guess FABind would still perform fine since (like DiffDock) it perceives receptors on the residue (amino acid) level. However, if the difference is also due to changes in the backbone, then I guess performance might suffer.

---

> > > ### Author Response · Authors · 2023-08-18
> > >
> > > Thanks for your acknowledgment and constructive comments.
> > >
> > > > **The importance of allosteric binding sites in drug discovery**
> > > >
> > >
> > > We agree with you regarding the significance of allosteric binding sites in drug discovery, which should be an interesting and important focus in the future. We will consider incorporating this perspective in future work.
> > >
> > > > **Explanation for FABind performing better than FABind + SMINA**
> > > >
> > >
> > > Sorry if any confusion we made. For this experiment, "FABind+SMINA" refers to the process where we first use FABind's pocket prediction module to predict the pocket, and then use SMINA to perform docking, while FABind uses both the prediction module and the docking module. Hence, the main difference lies in the docking module (one is SMINA, one is FABind). We attribute the superior performance of FABind to two main factors: the specially-designed structure of our docking module and the joint optimization which results in the two parts of FABind-pocket prediction and docking-being more compatible, providing mutual assistance.
> > >
> > > > **Apo-structure docking**
> > > >
> > >
> > > We have indeed observed discrepancies in both the backbone and side chains between the protein holo-structures and the aligned apo-structures, as the ESMFold itself introduces some errors. As you rightly pointed out, our FABind, as well as DiffDock, uses only the $C_\alpha$ of the main chain for modeling. The differences observed between our results for holo-structure docking and apo-structure docking further validate the inaccuracies in ESMFold's prediction of the main chain.
> > >
> > > We recognize that with the advancement of protein folding methods, there is an increasing need to perform docking on apo-structures. In light of this, we are considering several improvements for future work:
> > >
> > > 1. Introducing the apo-structure during training as a form of data augmentation. This would allow the model to better generalize to apo-structures.
> > > 2. Modeling the protein in a flexible way, allowing its structure to change during docking. This approach would be more versatile, accommodating both holo- and apo-structures through structural adjustments.
> > >
> > > We believe these improvements will enhance the robustness and accuracy of docking and are worth more in-depth exploration.

---

### Author Rebuttal · Authors · 2023-08-09

Dear Reviewers,

We would like to extend our heartfelt gratitude to each one of you for the time, effort, and expertise you have invested in reviewing our paper. Your insightful comments and constructive feedback have been instrumental in enhancing the quality of the research.

In response to your comments and to provide further clarity on some of the points raised, we have prepared additional results shown in figures and tables. Please find the uploaded PDF file and refer to the specific figures and tables as needed.

Once again, thank you for your valuable contributions to this work.

Sincerely,

Authors

---

### Decision · Program_Chairs · 2023-09-21

**Decision:**

Accept (poster)

**Comment:**

All reviewers acknowledge the comparable performance of the proposed method with recent sota on protein-ligand binding while being computationally more efficient. The improvement is due to a novel combination of existing methods as well as a few new tricks. The AC agrees with the unanimous ratings to accept as the paper has clear merits for an important task and thus should be disseminated.